

# Dual-polarization radar rainfall estimation in Korea according to raindrop shapes using a 2D Video Disdrometer

H. -L. Kim[1*], M.-K. Suk[1], H. -S. Park[2], G. -W. Lee[3], and J. -S. Ko[1]

[1]{Radar Analysis Division, Weather Radar Center, Korea Meteorological Administration, Seoul, South Korea}

[2]{Satellite Analysis Division, National Meteorological Satellite Center, Korea Meteorological Administration, Seoul, South Korea}

[3]{Department of Atmospheric Sciences, Kyungpook National University, Daegu, South Korea}

Correspondence to: H. -L. Kim (hlk0919@korea.kr)

## Abstract

The shapes of raindrops play an important role in inducing polarimetric rainfall algorithms with differential reflectivity ($Z_{DR}$) and specific differential phase ($K_{DP}$). The shapes of raindrops have a direct impact on rainfall estimation. However, the characteristics of raindrop size distribution (DSD) are different depending on precipitation type, storm stage of development, and regional and climatological conditions. Therefore, it is necessary to provide assumptions based on raindrop shapes that reflect the rainfall characteristics of the Korean peninsula. In this study, we presented a method to find optimal polarimetric rainfall algorithms on the Korean peninsula using the 2-Dimensional Video Disdrometer (2DVD) and Bislsan S-Band dual-polarization radar. First, a new axis ratio of raindrop relations was developed for the improvement of rainfall estimation. Second, polarimetric rainfall algorithms were derived using different axis ratio relations, and estimated radar-point one-hour rain rate for the differences in polarimetric rainfall algorithms were compared with the hourly rain rate measured by gauge. In addition, radar rainfall estimation was investigated in relation to calibration bias of reflectivity and differential reflectivity. The derived raindrop axis ratio relation from the 2DVD was more oblate than existing relations in the D < 1.5 mm



and D > 5.5 mm range. The R(K$_{DP}$, Z$_{DR}$) algorithm based on a new axis ratio relation showed
the best result on DSD statistics; however, the R(Z$_h$, Z$_{DR}$) algorithm showed the best
performance for radar rainfall estimation, because the rainfall events used in the analysis
were mainly weak precipitation and K$_{DP}$ is noisy at lower rain rates ($\leq$ 5 mm hr$^{-1}$). Thus, the
R(K$_{DP}$, Z$_{DR}$) algorithm is suitable for heavy rainfall and R(Z$_h$, Z$_{DR}$) algorithm is suited for
light rainfall. The calibration bias of reflectivity (Z$_H$) and differential reflectivity (Z$_{DR}$) were
calculated from the comparison of measured with simulated Z$_H$ and Z$_{DR}$ from the 2DVD. The
calculated Z$_H$ and Z$_{DR}$ bias was used to reduce radar bias, and to produce more accurate
rainfall estimation.

## 1    Introduction

Radar is a very useful monitoring tool for extreme weather forecasting, flood forecast
and rainfall estimation because of its high spatial and temporal resolution. In particular, dual-
polarization radar providing reflectivity (Z$_H$), differential reflectivity (Z$_{DR}$), differential phase
($\Phi_{DP}$), specific differential phase (K$_{DP}$), and cross-correlation coefficient ($\rho_{hv}$) can estimate
rainfall more accurately than single polarization radar. Dual-polarization radar provides
characteristics of the precipitation by backscatter and differential propagation phase of
hydrometeors, and therefore can obtain more information about DSD (Cifelli et al., 2011). In
addition, the multi-parameters can distinguish precipitation type, and reduce the impact of
DSD variability on the rainfall estimation. Therefore, rainfall estimators using a combination
of Z$_H$, Z$_{DR}$, and K$_{DP}$ are better than using reflectivity factor only (Ryzhkov et al., 2005).
Several different polarimetric rainfall algorithms have been developed assuming
raindrop shapes (Sachidananda and Zrnić, 1987; Chandrasekar et al., 1990; Ryzhkov and
Zrnic, 1995; Gorgucci et al., 2001). This is because the shape of raindrops is one of the most
sensitive parameters for representing the DSD properties of the rain. Some researchers have
attempted to produce the mean shape of raindrops. Keenan et al. (2001) derived an empirical
relation from observational data, and Pruppacher and Beard (1970), Green (1975), and Beard
and Chuang (1987) investigated the shape of raindrops falling under the influence of gravity.
The raindrop shape is defined by the shape-size relationship of a raindrop. These raindrop
axis-ratio relations play an important role in deriving polarimetric rainfall algorithms that use
Z$_{DR}$ and K$_{DP}$ (Jameson 1983, 1985; Gorgucci et al. 2001). However, the characteristics of rain
DSDs are associated with types of storms and stages of storm development as well as





climatic regimes (Bringi et al., 2003). Thus, it was necessary to determine the mean axis ratio
of raindrops reflecting rainfall characteristic of the Korean peninsula, to optimize the
polarimetric rainfall algorithm.

4        Polarization radar contains errors such as attenuation, bright band, ground clutter, and

calibration bias (of $Z_H$ and $Z_{DR}$). These measurement errors affect rainfall estimation.
Therefore, accurate measurement and calibration of $Z_H$ and $Z_{DR}$ are necessary to achieve
accurate radar rainfall estimation (Park and Lee, 2010). The accommodation of calibration
bias of single polarimetric radar is possible by monitoring the stability of the hardware, and
measured $Z_H$ and $Z_{DR}$ can be corrected using ground validation equipment such as a
disdrometer. Joss et al. (1968) calibrated radar reflectivity using the measured $Z_H$ from the
radar profiler (at vertical incidence) and disdrometer-inferred $Z_H$. The radar reflectivity was
calibrated by comparing reflectivity between radar and disdrometer to check the calibration
of the WSR-88D (Weather Surveillance Radars—1988 Doppler) at Greer, South Carolina
(Ulbrich and Lee, 1999). Goddard et al. (1982) and Goddard and Cherry (1984) compared
radar with disdrometer using the axis-ratio relations of Pruppacher and Beard (1970) and
Pruppacher and Pitter (1971), respectively, and found that radar measures of $Z_{DR}$ were 0.3 dB
and 0.1 dB lower than the disdrometer estimates. In addition to use of disdrometers, there are
various other ways to correct biases in radar data, such as using the self-consistency
constraint between $Z_H$ and $K_{DP}$, vertically pointing measurements, and comparison of
measured data and mean $Z_H$-$Z_{DR}$ relationship (Kwon et al., 2015). Moreover, a variety of
radar calibration methods were introduced in Atlas (2002).

22       In this study, we developed mean axis ratio relation and polarimetric rainfall

algorithms using 2-Dimensional Video Disdrometer (2DVD) measurement from September
2011 to October 2012 in Daegu, Korea. The four raindrop shapes assumption [after
Pruppacher and Beard (1970), Beard and Chuang (1987), and Brandes et al. (2002)] and
newly derived axis-ratio relation from 2DVD data were used to derive accurate polarimetric
rainfall relation for rainfall estimation. In addition, the $Z_H$ and $Z_{DR}$ of Bislan dual-
polarization radar were calibrated by comparing them with simulated $Z_H$ and $Z_{DR}$ by 2DVD.
Thereafter, improvement of quantitative rainfall estimation was investigated by applying
derived calibration bias. In Section 2, the data used in this study is described. The
methodology for 2DVD data quality control, derived raindrop-axis ratio from 2DVD data,
and simulated polarimetric parameters by the T-matrix scattering method are described in



Section 3. The results of the statistical validation of rainfall estimation are presented in
Section 4. Finally, conclusions are drawn in Section 5.
**2  Data and instrument**
**2.1  Disdrometer**
The disdrometer data was used for development of mean raindrop axis ratio and
polarimetric rainfall relations. The disdrometer data used in this study were measured using a
2DVD, and the data were collected from September 2011 to October 2012 in Daegu, Korea
(35.9°N, 128.5°E). The 2DVD consists of two orthogonal light sheets (referred to as A and B
line-scan cameras). Line-scan cameras have single-line photo detectors. The particle shadows
are detected on the photo detectors and the particle images are recoded from two sides and at
different heights, when particles are falling through the measurement area (10 cm × 10 cm). A
more detailed description of 2DVD is given in Kruger and Krajewski (2002).
The 2DVD measures drop size, fall velocity, and the shape of individual particles.
From these, one can calculate precipitation DSDs including such as the rain rate, total number
concentration, and water content.
**2.2  Radar**
The Ministry of Land, Infrastructure, and Transport (MLIT) operates the Bislsan (BSL)
dual-polarization radar in Bislsan, Korea (35.7°N, 128.5°E). The BSL S-Band radar has a
narrow observation range of 150 km and frequency of 2.5 min because it is used primarily for
hydrological observations and flood forecasts. The BSL S-Band radar measured polarimetric
parameters such as $Z_H$, $Z_{DR}$, $K_{DP}$ and $\rho_{hv}$ in real time. The data obtained were from six
elevation angles (from −0.5° to 1.6°), with a gate size resolution of 125 m and radar beam
width of 0.95°. The specifications of the BSL radar are shown in Table 1.
The BSL S-Band dual-polarization radar is located about 22.3 km (17°) away from the
2DVD location (Fig. 1). These geographical locations were adopted to compare the two sets
of observation data. We used radar data from September 2011 to October 2012. During this
period, rainfall events were analyzed for products of calibration bias of $Z_H$ and $Z_{DR}$ and





rainfall estimation. In addition, the 0.0° plan position indicator (PPI) radar data were used to
avoid effects from beam blocking and ground echoes on the measurements. The $Z_H$, $Z_{DR}$, $\Phi_{DP}$,
and $\rho_{hv}$ radar parameters were averaged over five successive gate size resolutions and two
adjacent azimuth angles, and $K_{DP}$ was calculated from the filtered $\Phi_{DP}$ as the slope of a least
squares fit.

## 2.3   Rain gauge

7        A tipping bucket rain gauge was used to validate rainfall calculated from 2DVD data.

The bucket size of the rain gauge was 0.2 mm and time resolution was 0.5 s. The rain gauge
was installed in the same location as the 2DVD.

## 3   Methodology

### 3.1   Quality control of 2DVD

13        The 2DVD observation data is useful to investigate the characteristics of rainfall.

However, a number of particle outliers were measured due to wind turbulence, splashing,
break up of drops, and mismatching between camera A and B (Raupach and Berne, 2015).
These results lead to incorrect information about the particles. Therefore, before using the
2DVD data, a quality control process was needed.

18        Figures 2a and b show fall velocity and oblateness distribution according to raindrop

diameter before data quality control was performed. For comparison, we plotted the axis-
ratio-diameter relation of Pruppacher and Beard (1970). Some of the outliers of fall velocity
and oblateness distribution were beyond the normal distribution. In particular, the outliers
appeared prominently in the small raindrop ranges. To remove these outliers, velocity-based
filtering was applied to the 2DVD measurement data (Thurai and Bringi, 2005).

$$|V_{measured}(D) - V_A(D)| < 0.4 V_A(D)$$

$$V_A(D) = 9.65 - 10.3\exp(-0.6D) \tag{1}$$

where, D [mm] is the drop diameter, $V_{measured}$ [ms$^{-1}$] is the fall velocity as measured by the
2DVD, and $V_A$ represents the Atlas velocity formula (Atlas et al. 1973). Despite the
application of velocity-based filtering, significant bias still remained in the small drop size
area. This was due to instrumental limitations, such as mismatch problems with the line-scan





cameras and the limited vertical resolution of the instrument. Therefore, the oblateness data
corresponding to raindrop diameters smaller than 0.5 mm were removed when we calculated
the new axis-ratio formula. The values outside the normal distribution (about 17%) were
removed as a result of application filtering (Fig. 2c).
To analyze the reliability of the 2DVD data, we compared the rain rate calculated
from the 2DVD data (Eq. 2) to collocated rain gauge measurements. The difference of
accumulated rainfall represents the percent error (Eq. 3).

$$R = 6 \times 10^{-4} \pi \sum_{D_{min}}^{D_{max}} D^3 V(D) N(D) \Delta D \quad [mm\ hr^{-1}] \qquad (2)$$

$$PE = \frac{|AR_{rain\ gauge} - AR_{2DVD}|}{AR_{rain\ gauge}} \times 100 \quad [\%] \qquad (3)$$

here $D_{max}$ and $D_{min}$ are the maximum and minimum diameters of the observed drops in mm,
N(D) is the drop number concentration in $mm^{-1}m^{-3}$, V(D) is the drop fall velocity in $ms^{-1}$, and
$\Delta D$ is drop interval ($\Delta D = 0.2$ mm). The drop fall velocity formula was derived by Brandes
et al. (2002); *PE* is the percent error, and *AR* is accumulated rainfall in mm.
During the period from September 2011 to October 2012, the rainfall cases were
analyzed and an example of six cases is shown in Fig. 3. The six events are (i) 0000–0900
UTC 14 October 2011, (ii) 1400–2359 UTC 2 April 2012, (iii) 0000–2359 UTC 21 April
2012, (iv) 0000–0800 UTC 25 April 2012, (v) 0000–2359 UTC 23 August 2012, and (vi)
1600–2359 UTC 27 August 2012. Figure 3a shows the accumulated rainfall computed from
the 2DVD and rain gauge on 14 October 2011. The overall distribution between the 2DVD
and rain gauge was good. The 2DVD recorded 13.14 mm and rain gauge recorded 13.52 mm,
their difference was about 2.81%. As shown in Fig. 3b–f, percent errors from the five rainfall
cases are 9.42, 0.24, 4.16, 7.88 and 8.25%, respectively. Generally, some papers show that
rainfall differences between disdrometer and rain gauge were mostly from 10% to 20%
(McFarquhar and List, 1993; Sheppard and Joe, 1994; Hagen and Yuter, 2003; Tokay et al.,
2003). These differences might result from such issues as differences in instruments, effects
from the measurement environment, and rainfall variability. Therefore, the 2DVD data within
20% percent error were used in this study.
After the quality control process, a total of 33 rainfall cases were selected for
investigating the characteristics of rainfall over the Korea Peninsula. The accuracy of 33





rainfall cases was in the range of 0.24–19.32% compared to in suit rain gauges. The dataset
consisted of 15 stratiform rainfall cases, 12 convective rainfall cases, and 6 mixed rainfall
cases (total of 33 rainfall event) with 17,618 min DSD samples. The type of precipitation,
difference rainfall, and accumulated rainfall between 2DVD and rain gauge for the 33 rainfall
events are listed in Table 2. Figure 4 shows hourly and total accumulated rainfall of 2DVD
and rain gauge for the 33 rainfall cases. The overall rainfall distribution between 2DVD and
rain gauge were good, total accumulated rainfall by rain gauge was larger than 2DVD by
about 0.81%.
**3.2   Raindrop axis ratio**

11       A very small raindrop has an approximately spherical shape that becomes oblate as its

size increases. The shape of a raindrop according to drop size can be expressed as the mean
axis-ratio relation; this relation is one of the most sensitive parameters for representing the
rainfall properties. Hence, in order to produce rainfall estimation algorithms reflecting
rainfall characteristic of the Korean peninsula, the new mean axis-ratio relation, using the
2DVD data listed in Table 2, was derived as a polynomial function. Although the measured
maximum diameter from the 2DVD could reach axis-about 8.0 mm, the mean axis-ratio
fitting was established to within 7 mm in order to obtain accurate information from the
appropriate data. The third-order polynomial new mean axis-ratio relation (b/a) is as follows
in Eq. (4), which is reasonably extended to 7 mm.
$b/a = 0.997845 - 0.0208475D - 0.0101085D^2 + 6.4332 \times 10^{-4}D^3$         $(0.5 \leq D \leq 7 \text{ mm})$   (4)
where, $a$ and $b$ are the major axis and minor axis, respectively. D is the raindrop diameter in
mm.
**3.3   Disdrometer-rainfall algorithms**

26       In order to produce the polarimetric rainfall algorithms, the theoretical polarimetric

parameters (e.g., $Z_H$, $Z_{DR}$, and $K_{DP}$) were simulated from the 2DVD data using the T
(transition) matrix method. Polarimetric parameters were simulated by making assumptions
about the shape of the raindrops. First, we calculated the complex scattering amplitudes of





raindrops at the S-Band of wavelength 10.7 cm using mean axis-ratio relations. Second,
calculated scattering amplitudes about the axis ratio relations were used for production of
polarimetric parameters. The dual-polarimetric parameters were calculated using the
following Eq. (5–7) (Jung et al., 2010).

$$Z_h = \frac{4\lambda^4}{\pi^4 |K_w|^2} \int_0^{D_{max,x}} A|f_a(\pi)|^2 + B|f_b(\pi)|^2 + 2CRe[f_a(\pi)f_b^*(\pi)]N(D)dD \ [mm^6 m^{-3}] \ (5)$$

$$Z_v = \frac{4\lambda^4}{\pi^4 |K_w|^2} \int_0^{D_{max,x}} B|f_a(\pi)|^2 + A|f_b(\pi)|^2 + 2CRe[f_a(\pi)f_b^*(\pi)]N(D)dD \ [mm^6 m^{-3}] \ (6)$$

Where
$A = <\cos^4\Phi> = \frac{1}{8}(3 + 4\cos2\bar{\emptyset}e^{-2\sigma^2} + \cos4\bar{\emptyset}e^{-8\sigma^2})$

$B = <\sin^4\Phi> = \frac{1}{8}(3 - 4\cos2\bar{\emptyset}e^{-2\sigma^2} + \cos4\bar{\emptyset}e^{-8\sigma^2})$

And

$C = <\sin^2\Phi\cos^2\Phi> = \frac{1}{8}(1 - \cos4\bar{\emptyset}e^{-8\sigma^2})$

$$K_{DP} = \frac{180\lambda}{\pi} \int_0^{D_{max}} C_k Re[f_a(0) - f_b(0)]N(D)dD \ [°km^{-1}] \qquad (7)$$

where $C_k = <\cos2\Phi> = \cos2\Phi e^{-2\sigma^2}$.
where, $f_a(0)$ and $f_b(0)$ are complex forward-scattering amplitudes, and $f_a(\pi)$ and $f_b(\pi)$ are
complex backscattering amplitudes for polarization along the major and minor axes. Here, $f_a^*$
and $f_b^*$ are their respective conjugates, $\bar{\emptyset}$ is mean canting angle, and σ is standard deviation of
the canting angle. The terms $\bar{\emptyset}$ and σ are assumed to be 7° and 0°, respectively (Huang et al.,
2008). $D_{max}$ is 7 mm, the radar wavelength is λ = 10.7 cm (S-Band), the dielectric factor for
water is $K_w = 0.93$, and N(D) was calculated using the 2DVD measurement.

15       Polarimetric rainfall relations between R and dual-polarimetric parameters are derived

when rain rate is greater than 0.1 mm hr[-1]. Derived new polarimetric rainfall relations
according to axis ratio relations are presented in Table3.



### 3.4  Calibration of radar
The polarimetric radar contains systematic bias of the radar itself. Therefore,
accommodation of the calibration bias of radar is necessary to improve quantitative
precipitation estimation. The calibration of the radar was done for light rainfall events, and
the new axis-ratio relation (Eq. 4) was used for simulation of the theoretical $Z_H$ and $Z_{DR}$
parameters.
The calibration bias of $Z_H$ and $Z_{DR}$ were calculated from the comparison of measured
$Z_H$ and $Z_{DR}$ with the simulated $Z_H$ and $Z_{DR}$ from the 2DVD measurement. To compare
polarimetric radar parameters, the cross-match point must first be determined. This is,
because 2DVD data are point measurements and radar data are volume measurements. The
BSL S-Band radar data were averaged over five successive gates and two adjacent azimuth
angles centered on the 2DVD location. The elevation angle of 0.0° PPI was used.
## 4  Results
### 4.1  Comparison of raindrop axis ratio relations
We compared the new axis-ratio experimental fit with existing mean axis-ratio
relations such as those of Pruppacher and Beard (1970), Beard and Chuang (1987), and
Brandes et al. (2002) in Fig. 5. These have been approximated to various polynomial
formulas, as follows:
$b/a = 1.03-0.062D$ $\qquad$ (1 ≤ D ≤ 9 mm) (8)
$b/a = 1.0048+5.7\times10^{-4}D-2.628\times10^{-2}D^2+3.682\times10^{-3}D^3-1.677\times10^{-4}D^4$ $\quad$ (1 ≤ D ≤ 7 mm) (9)
$b/a = 0.9951+0.02510D-0.03644D^2+5.303\times10^{-3}D^3-2.492\times10^{-4}D^4$ $\quad$ (1 ≤ D ≤ 8 mm) (10)
Equation 8 from Pruppacher and Beard (1970) is a linear relation from wind tunnel data, Eq.
(9) is a fourth-order polynomial formula to the numerical model. Equation 10 is a polynomial
empirical relation developed by Brandes et al. (2002) that was derived by combining drop
shape observations. The Pruppacher and Beard (1970) linear relation (Eq. (8), green dash-dot
line) falls below the new mean axis-ratio for 1 ≤ D < 5 mm, and the Beard and Chuang (1987)
polynomial relation (Eq. (9), black dashed line) is slightly lower than the result of the new
mean axis-ratio in the range 2.5–6.5 mm. The new mean axis-ratio fit is more oblate than the





Brandes et al. (2002) polynomial empirical relation (Eq. (10), blue dotted line) when the
raindrop sizes are greater than 5.5 mm and less than 2.5 mm. To the exclusion of this part, the
new axis ratio of raindrops in the range 3–5.5 mm was similar to Eq. (10).

4        Thus, the new mean axis ratio relation is very similar to existing axis-ratio relations

except for small particles ($\leq 2$ mm) and large particles ($\geq 5.5$ mm). This means that raindrops
in South Korea are more oblate than the others. Although the difference in the axis ratio
seems small, its impact on the rainfall estimation cannot be neglected.

## 4.2  Verification of polarimetric rainfall algorithms

### 4.2.1  Variability of DSD in rainfall estimation

11       To investigate the variability of DSD in rainfall estimation from polarimetric

parameters, rain rate $R_e$ was estimated from various combinations of polarimetric variables
and compared with R from Eq. (2). The mean absolute error (MAE), the root-mean-square
error (RMSE), and correlation coefficient (Corr.) are defined by the following equation:

$$MAE = \frac{1}{N}\sum |R - R_e| \quad [mm\ h^{-1}] \tag{11}$$

$$RMSE = \left(\frac{1}{N}\sum (R - R_e)^2\right)^{0.5} \quad [mm\ h^{-1}] \tag{12}$$

$$\mathrm{Corr.} = \frac{1}{N-1}\frac{\sum[(R - \bar{R})(R_e - \overline{R_e})]}{\sqrt{Var(R)Var(R_e)}} \tag{13}$$

where R is rain rate from observed 1-min DSDs and $R_e$ is rain rate from estimated various
combination of polarimetric parameters. $R_e$ is then obtained from the same dataset. The N is
the number of comparisons. Figure 6 shows the scatterplot of R and $R_e$ for polarimetric
rainfall relations based on the new axis-ratio relation and the scatter indicates the effect of
DSD variability on rain estimation. The comparison between rain rates observed R and those
estimated $R_e$ shows good overall agreement. In particular, the statistic of scatter showed the
best result (MAE = 0.23 mm hr$^{-1}$, RMSE = 0.35 mm h$^{-1}$ and Corr = 0.10) when using the
R($K_{DP}$, $Z_{DR}$) based on the new axis-ratio relation. The use of the single parameter R($Z_H$)
results in increase of the MAE = 0.96 mm hr$^{-1}$ and RMSE = 2.40 mm hr$^{-1}$, and decrease of the
Corr = 0.93 when compared with other polarimetric rainfall relations. Other polarimetric



rainfall algorithms based on mean axis-ratio relations showed similar results. Thus, the
polarimetric parameters with $K_{DP}$ and $Z_{DR}$ from dual-polarization radar reduce the DSD
variability in the rainfall estimation. A summary of statistics according to polarimetric
rainfall algorithms and mean axis-ratio relations are presented in Table 3.
### 4.2.2  Validation of rainfall estimation

7        In order to evaluate radar rainfall estimation according to different rainfall relations

and raindrop shapes, we compared an estimated one-hour rain rate from the BSL S-Band
radar to the hourly rain rate by rain gauge in Daegu, Korea. In addition, the rainfall estimate
from the 2DVD was included for comparison. Statistical validation of the radar and 2DVD
rainfall estimates for the different rainfall relations were performed for 18 rainfall events
among the 33 rainfall cases. The mean absolute error (MAE) and the root-mean-square error
(RMSE) are given by

$$MAE = \frac{1}{N}\sum |R_R - R_G| \;\; [mm\ h^{-1}] \tag{12}$$

$$RMSE = \left(\frac{1}{N}\sum (R_R - R_G)^2\right)^{0.5} \;\; [mm\ h^{-1}] \tag{13}$$

where, $R_R$ is the averaged one-hour rain rate [mm h$^{-1}$] for the radar (or 2DVD), and $R_G$ is the
averaged one-hour rain rate [mm h$^{-1}$] for the rain gauge. The results are presented in Table 4.

16       Rainfall estimation from the 2DVD showed good results in the following order:

$R(K_{DP}, Z_{DR}) > R(Z_h, Z_{DR}) > R(K_{DP}) > R(Z_h)$. According to the DSD statistics, the combined
polarimetric rainfall relations using $Z_{DR}$ and $K_{DP}$ performed better than the single rainfall
relation for estimated rainfall. As can be seen from Table 4, $R(K_{DP}, Z_{DR})$ based on the new
axis-ratio relation performed better on DSD statistics, with MAE = 0.61 mm hr$^{-1}$, and RMSE
= 0.86 mm hr$^{-1}$. However, the $R(K_{DP}, Z_{DR})$ algorithm showed the worst results for radar
rainfall estimation, and the $R(Z_h, Z_{DR})$ algorithm showed the best performance. These results
can be found in Fig. 7, which shows a scatterplot of the one-hour rain rate from the rain
gauge and the radar (or 2DVD) data, using the new axis-ratio relation for 18 rainfall events.
The plus represents one-hour gauge rain rate versus radar rain rate from different rainfall
relations and the square indicates gauge versus 2DVD rain rate.



According to the DSD results, $K_{DP}$ is less sensitive to DSD variation and uncertainties
in raindrop shapes; however, the accuracy of the rainfall estimation declined when the $K_{DP}$
parameter was used for radar rainfall estimation. Moreover, the radar rainfall estimations
from $R(K_{DP})$ and $R(K_{DP}, Z_{DR})$ exceeded rainfall gauge measurements at lower rain rates ($\leq 5$
mm hr$^{-1}$), whereas rainfall estimations from $R(Z_h, Z_{DR})$ were similar to rainfall by measured
by gauges. In addition, the radar rainfall estimations from $R(K_{DP})$ and $R(K_{DP}, Z_{DR})$ perform
better than those of $R(Z_h, Z_{DR})$ for rain rates exceeding 5 mm hr$^{-1}$. In other words, as the rain
rate increased, the uncertainty of $K_{DP}$ from the radar declined. This was because $K_{DP}$ is noisy
in light rainfall. Thus, the $R(K_{DP}, Z_{DR})$ relation is best used for heavy rainfall and $R(Z_h, Z_{DR})$
is suited for light rainfall.
In addition, the polarimetric rainfall relations based on the new axis-ratio relation also
were better than the others. Although the difference in the value of the statistics seems small
according to raindrop axis ratio relations, it has an effect on the accuracy of rainfall
estimation. Therefore, rainfall characteristics should be reflected in polarimetric rainfall
relations.
**4.2.3  Correction of calibration bias**
In this study, the calibration bias of $Z_H$ and $Z_{DR}$ was calculated for eight rainfall events,
and the $R(Z_h, Z_{DR})$ algorithm based on Eq. (4) was used to estimate rainfall. Figure 8 and 9
shows comparison of the time series and scatter diagrams of $Z_H$ and $Z_{DR}$ for the 2DVD and
BSL radar. The overall distribution of the observed $Z_H$ corresponded well with the simulated
parameter; however, the measured $Z_{DR}$ at BSL was underestimated compared to the simulated
$Z_{DR}$ value. The mean bias (= bias) of $Z_H$ and $Z_{DR}$ on 14 May 2012 was about 2.17 dBZ and
0.28 dB, respectively (Fig. 8). The bias of $Z_H$ and $Z_{DR}$ on 23 August 2012 was 0.98 dBZ and
0.10 dB (Fig. 9).
The accuracy of the radar rainfall estimation was investigated by applying calculated
$Z_H$ and $Z_{DR}$ bias. These results were evaluated by comparing with rain gauge measurements.
Figure 10a shows the one-hour rain rate (left ordinate) and accumulated rainfall (right
ordinate) estimated from the radar and rain gauge on May 14, 2012. The blue and green solid
lines are estimated one-hour rainfall rates from before and after bias correction, and the bar
graph is the one-hour rainfall rate measured by rain gauge. The blue and green dotted lines



are estimated accumulated rainfall from before and after bias correction, and the red dotted
line is the accumulated rainfall by the rain gauge. In comparison with the rain gauge
measurement, underestimated precipitation (12.67 mm) was corrected to 15.33 mm after bias
correction. When the estimated rainfall was compared to the rain gauge as ground truth,
rainfall estimation was improved about 13.71%. Figure 10b shows the results for the period
00:00 to 23:59 UTC on August 23, 2012. The accumulated rainfall recorded was 83.86, 71.52,
and 80.12 mm for the rain gauge, before bias correction, and after bias correction. Rainfall
estimation was improved about 10.25%. For eight rainfall events, the total mean bias of $Z_H$
and $Z_{DR}$ from the BSL radar was 1.03 dBZ and 0.22 dB, respectively. Moreover, MAE fell by
1.03 to 0.93 mm hr$^{-1}$ and RMSE decreased by 1.41 to 1.26 mm hr$^{-1}$. The bias of $Z_H$ and $Z_{DR}$,
MAE and RMSE results for each of the eight rainfall events are presented in Table 5. As
shown in Table 5, rainfall estimation tended to improve after bias correction.
**5   Conclusion**
The purpose of this study was to find an optimal polarimetric rainfall algorithm using
2DVD measurement in Korea, and to improve rainfall estimation by correcting $Z_H$ and $Z_{DR}$
calibration bias. First, we derived a new raindrop axis-ratio relation reflecting rainfall
characteristics on the Korean peninsula, using data from 33 rainfall events, after checking the
accuracy and quality control of the 2DVD measurements. The derived raindrop axis-ratio
relation was compared with existing relations. Although the difference in relations seems
small, its impact on the polarimetric rainfall algorithm cannot be neglected.
The polarimetric rainfall relations were derived based on various assumptions about the
shape of raindrops, and the accuracy validation of one-hour rainfall rate for rainfall
algorithms was conducted using 2DVD, BSL radar, and rain gauge. As a result, R($K_{DP}$, $Z_{DR}$)
based on the new axis-ratio relation was suited for rainfall estimation of the DSD statistic
when compared with others. However, the $K_{DP}$-based algorithms had a large statistical error
for radar rainfall estimation, and R($Z_h$, $Z_{DR}$) based on the new axis-ratio relation showed the
best performance on BSL S-Band radar rainfall estimation. This was because the measured
$K_{DP}$ parameter was weak at lower rain rates ($\leq$ 5 mm hr$^{-1}$). To calculate the calibration bias of
radar, measured $Z_H$ and $Z_{DR}$ were compared with those simulated. Calculated $Z_H$ and $Z_{DR}$ bias
was used to reduce radar bias, and to produce more accurate rainfall estimation. After bias
correction, rainfall estimated from radar was close to that measured using the rain gauge.



In this paper, different axis ratios of raindrops were used to derive new polarimetric
rainfall relations, and the new polarimetric rainfall algorithms were assessed for point radar
rainfall estimation. The effect of areal rainfall estimation and classification of rain rate on
polarimetric rainfall relations will be studied in future work.



## Acknowledgements

This research was supported by the "Development and application of cross governmental dual-pol radar harmonization (WRC-2013-A-1)" project of the Weather Radar Center, Korea Meteorological Administration.

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



1    Table 1. Specification of dual-polarization radar in Bislsan.

| Parameters | | Characteristics |
|---|---|---|
| Variables | | $Z_H$, $V_r$, SW, $Z_{DR}$, $\Phi_{DP}$, $K_{DP}$, $\rho_{hv}$ |
| Altitude of radar antenna | | 1085 m |
| Transmitter type | | Klystron |
| Transmitter peak power | | 750 kW |
| Antenna diameter | | 8.5 m |
| Beam width of radar | | 0.95° |
| Observation | Frequency | 2,785 MHz (S-band) |
| | Range | 150 km |
| | Gate size | 125 m |
| | Elevations | -0.5°, 0°, 0.5°, 0.8°, 1.2°, 1.6° (6 elevation) |



1     Table 2. Summary of the date, type of precipitation, and accumulated rainfall comparison

2     between 2DVD and rain gauge for the 33 rainfall events.

| Date | Time of observation [UTC] | Type of Precipitation | Accumulated rainfall [mm] | | PE [%] |
|---|---|---|---|---|---|
| | | | 2DVD | Rain gauge | |
| `11.09.05 | 01:00-15:00 | C | 7.07 | 6.86 | 3.12 |
| `11.09.09 | 15:20-21:20 | S | 4.26 | 4.31 | 1.17 |
| `11.09.10 | 00:00-23:59 | C | 19.50 | 17.83 | 9.37 |
| `11.09.29 | 00:00-17:00 | S | 3.81 | 3.72 | 2.38 |
| `11.10.13 | 00:00-23:59 | S | 3.91 | 4.11 | 4.98 |
| `11.10.14 | 00:00-09:00 | S | 13.14 | 13.52 | 2.81 |
| `11.10.21 | 06:00-23:59 | S | 53.11 | 65.83 | 19.32 |
| `12.04.02 | 14:00-23:59 | M | 17.79 | 16.26 | 9.42 |
| `12.04.21 | 00:00-23:59 | S | 32.06 | 32.13 | 0.24 |
| `12.04.25 | 00:00-08:00 | S | 18.57 | 17.83 | 4.16 |
| `12.05.01 | 08:00-21:00 | S | 4.07 | 3.53 | 15.32 |
| `12.05.08 | 07:00-11:00 | C | 8.98 | 9.60 | 6.50 |
| `12.05.14 | 00:00-15:00 | S | 22.19 | 19.98 | 11.03 |
| `12.05.28 | 06:00-07:00 | C | 15.26 | 14.50 | 5.22 |
| `12.06.08 | 03:00-23:59 | C | 17.09 | 19.40 | 11.91 |
| `12.06.23 | 00:00-08:00 | C | 13.48 | 14.50 | 7.05 |
| `12.07.06 | 00:00-18:00 | C | 22.40 | 20.18 | 10.98 |
| `12.07.12 | 17:00-21:00 | M | 7.49 | 7.45 | 0.58 |
| `12.07.13 | 01:00-12:00 | C | 25.26 | 25.08 | 0.71 |
| `12.07.15 | 00:00-12:00 | C | 8.22 | 7.64 | 7.55 |
| `12.07.16 | 15:00-23:59 | M | 16.33 | 18.42 | 11.31 |
| `12.07.21 | 09:00-10:30 | C | 5.85 | 5.68 | 2.99 |
| `12.08.12 | 01:00-18:00 | C | 20.27 | 18.42 | 10.08 |
| `12.08.13 | 00:00-15:00 | C | 37.19 | 34.09 | 9.10 |
| `12.08.23 | 00:00-23:59 | M | 90.47 | 83.86 | 7.88 |
| `12.08.24 | 00:00-23:59 | S | 7.32 | 7.45 | 1.64 |
| `12.08.27 | 16:00-23:59 | S | 13.57 | 12.54 | 8.25 |
| `12.08.29 | 19:00-23:59 | S | 6.23 | 5.29 | 17.72 |
| `12.09.09 | 00:00-23:59 | S | 21.45 | 18.65 | 15.04 |
| `12.09.16 | 00:00-23:59 | S | 82.68 | 69.88 | 18.32 |
| `12.09.17 | 00:00-07:00 | M | 63.51 | 58.55 | 8.47 |
| `12.10.22 | 06:00-11:00 | M | 13.60 | 14.92 | 8.81 |
| `12.10.27 | 00:00-09:00 | S | 8.74 | 8.20 | 6.53 |



1    Table 3. List of different polarimetric rainfall relations used for rainfall estimation and MAE,

2    RMSE and correlation coefficient for estimated rain rate vs observations.

| $R(Z_h)=\alpha|Z_h|^{\beta}$ | | | | | |
|---|---|---|---|---|---|
| Polarimetric rainfall relation | | Scatterplot R-R$_e$ | | | Assumptions |
| $\alpha$ | $\beta$ | MAE | RMSE | Corr. | |
| 1   0.0558 | 0.5894 | 0.96 | 2.39 | 0.93 | Pruppacher and Beard (1970) |
| 2   0.0576 | 0.5867 | 0.96 | 2.41 | 0.92 | Beard and Chuang (1987) |
| 3   0.0577 | 0.5871 | 0.96 | 2.41 | 0.92 | Brandes et al. (2002) |
| 4   0.0565 | 0.5889 | 0.96 | 2.40 | 0.93 | New axis ratio (Experimental fit) |
| $R(K_{DP})=\alpha|K_{DP}|^{\beta}$ | | | | | |
| Polarimetric rainfall relation | | Scatterplot R-R$_e$ | | | Assumptions |
| A | B | MAE | RMSE | Corr. | |
| 1   38.66 | 0.837 | 0.46 | 1.04 | 0.99 | Pruppacher and Beard (1970) |
| 2   43.77 | 0.768 | 0.66 | 1.42 | 0.97 | Beard and Chuang (1987) |
| 3   46.97 | 0.743 | 0.82 | 1.63 | 0.97 | Brandes et al. (2002) |
| 4   42.28 | 0.833 | 0.45 | 1.14 | 0.98 | New axis ratio (Experimental fit) |
| $R(Z_h, Z_{DR})=\alpha Z_h^{\beta}10^{0.1\gamma ZDR}$ | | | | | |
| Polarimetric rainfall relation | | Scatterplot R-R$_e$ | | | Assumptions |
| $\alpha$ | B | $\gamma$ | MAE | RMSE | Corr. |
| 1   0.0110 | 0.89 | -4.0808 | 0.45 | 0.77 | 0.99 | Pruppacher and Beard (1970) |
| 2   0.0091 | 0.88 | -3.5197 | 0.46 | 0.83 | 0.99 | Beard and Chuang (1987) |
| 3   0.0088 | 0.88 | -3.4789 | 0.47 | 0.85 | 0.99 | Brandes et al. (2002) |
| 4   0.0112 | 0.87 | -3.7613 | 0.48 | 0.89 | 0.99 | New axis ratio (Experimental fit) |
| $R(K_{DP}, Z_{DR})=\alpha K_{DP}^{\beta}10^{0.1\gamma ZDR}$ | | | | | |
| Polarimetric rainfall relation | | Scatterplot R-R$_e$ | | | Assumptions |
| $\alpha$ | $\beta$ | $\gamma$ | MAE | RMSE | Corr. |
| 1   66.23 | 0.96 | -1.3859 | 0.29 | 0.44 | 0.10 | Pruppacher and Beard (1970) |
| 2   83.78 | 0.93 | -1.6703 | 0.44 | 0.66 | 0.10 | Beard and Chuang (1987) |
| 3   96.37 | 0.92 | -1.8938 | 0.58 | 0.85 | 0.99 | Brandes et al. (2002) |
| 4   74.54 | 0.97 | -1.5328 | 0.23 | 0.35 | 0.10 | New axis ratio (Experimental fit) |





1    Table 4. Mean absolute error and root mean square error of the radar estimates of hourly rain

2    rate for the different radar rainfall algorithms listed in Table 3.

| | | | | | |
|---|---|---|---|---|---|
| **$R(Z_H)=\alpha|Z_H|^{\beta}$** | | | | | |
| | MAE | | RMSE | | Assumptions |
| | RADAR | 2DVD | RADAR | 2DVD | |
| 1 | 1.02 | 0.95 | 1.38 | 1.23 | Pruppacher and Beard (1970) |
| 2 | 1.03 | 0.96 | 1.39 | 1.24 | Beard and Chuang (1987) |
| 3 | 1.03 | 0.96 | 1.39 | 1.24 | Brandes et al. (2002) |
| 4 | 1.02 | 0.95 | 1.39 | 1.23 | New axis ratio (Experimental fit) |
| **$R(K_{DP})=\alpha|K_{DP}|^{\beta}$** | | | | | |
| | MAE | | RMSE | | Assumptions |
| | RADAR | 2DVD | RADAR | 2DVD | |
| 1 | 6.04 | 0.68 | 6.98 | 0.92 | Pruppacher and Beard (1970) |
| 2 | 7.69 | 0.74 | 8.63 | 0.99 | Beard and Chuang (1987) |
| 3 | 8.67 | 0.75 | 9.62 | 1.00 | Brandes et al. (2002) |
| 4 | 6.80 | 0.70 | 7.79 | 0.92 | New axis ratio (Experimental fit) |
| **$R(Z_H, Z_{DR})=\alpha Z_H^{\beta}10^{0.1\gamma ZDR}$** | | | | | |
| | MAE | | RMSE | | Assumptions |
| | RADAR | 2DVD | RADAR | 2DVD | |
| 1 | 0.88 | 0.65 | 1.20 | 0.90 | Pruppacher and Beard (1970) |
| 2 | 0.86 | 0.67 | 1.19 | 0.93 | Beard and Chuang (1987) |
| 3 | 0.87 | 0.71 | 1.21 | 0.99 | Brandes et al. (2002) |
| 4 | 0.84 | 0.71 | 1.17 | 0.97 | New axis ratio (Experimental fit) |
| **$R(K_{DP}, Z_{DR})=\alpha K_{DP}^{\beta}10^{0.1\gamma ZDR}$** | | | | | |
| | MAE | | RMSE | | Assumptions |
| | RADAR | 2DVD | RADAR | 2DVD | |
| 1 | 8.56 | 0.62 | 10.03 | 0.88 | Pruppacher and Beard (1970) |
| 2 | 11.33 | 0.65 | 13.01 | 0.91 | Beard and Chuang (1987) |
| 3 | 13.16 | 0.68 | 15.01 | 0.94 | Brandes et al. (2002) |
| 4 | 9.61 | 0.61 | 11.23 | 0.86 | New axis ratio (Experimental fit) |





1    Table 5. Mean absolute error and root mean square error of rainfall estimates before and after

2    applying bias correction.

|   | Date | Type of precipitation | Calibration Bias | | MAE [mm hr$^{-1}$] | | RMSE [mm hr$^{-1}$] | |
|---|---|---|---|---|---|---|---|---|
|   |   |   | $Z_h$ [dBZ] | $Z_{DR}$ [dB] | Before BC | After BC | Before BC | After BC |
| 1 | `11.10.13 | S | -0.01 | 0.10 | 0.48 | 0.38 | 0.59 | 0.47 |
| 2 | 10.21 | S | 1.16 | 0.28 | 0.75 | 0.74 | 0.98 | 0.97 |
| 3 | `12.04.25 | S | 1.40 | 0.43 | 1.31 | 1.26 | 2.00 | 1.89 |
| 4 | 05.14 | S | 2.17 | 0.28 | 0.58 | 0.43 | 0.80 | 0.59 |
| 5 | 08.23 | M | 0.98 | 0.10 | 0.80 | 0.64 | 1.33 | 0.91 |
| 6 | 09.09 | S | 0.44 | 0.18 | 0.73 | 0.67 | 1.16 | 1.07 |
| 7 | 09.16 | S | 0.90 | -0.13 | 0.77 | 0.71 | 0.95 | 0.92 |
| 8 | 10.22 | M | -1.44 | -0.14 | 2.83 | 2.58 | 3.47 | 3.29 |
| Avg |   |   | 1.03 | 0.22 | 1.03 | 0.93 | 1.41 | 1.26 |

3                                                                 * BC : Bias Correction





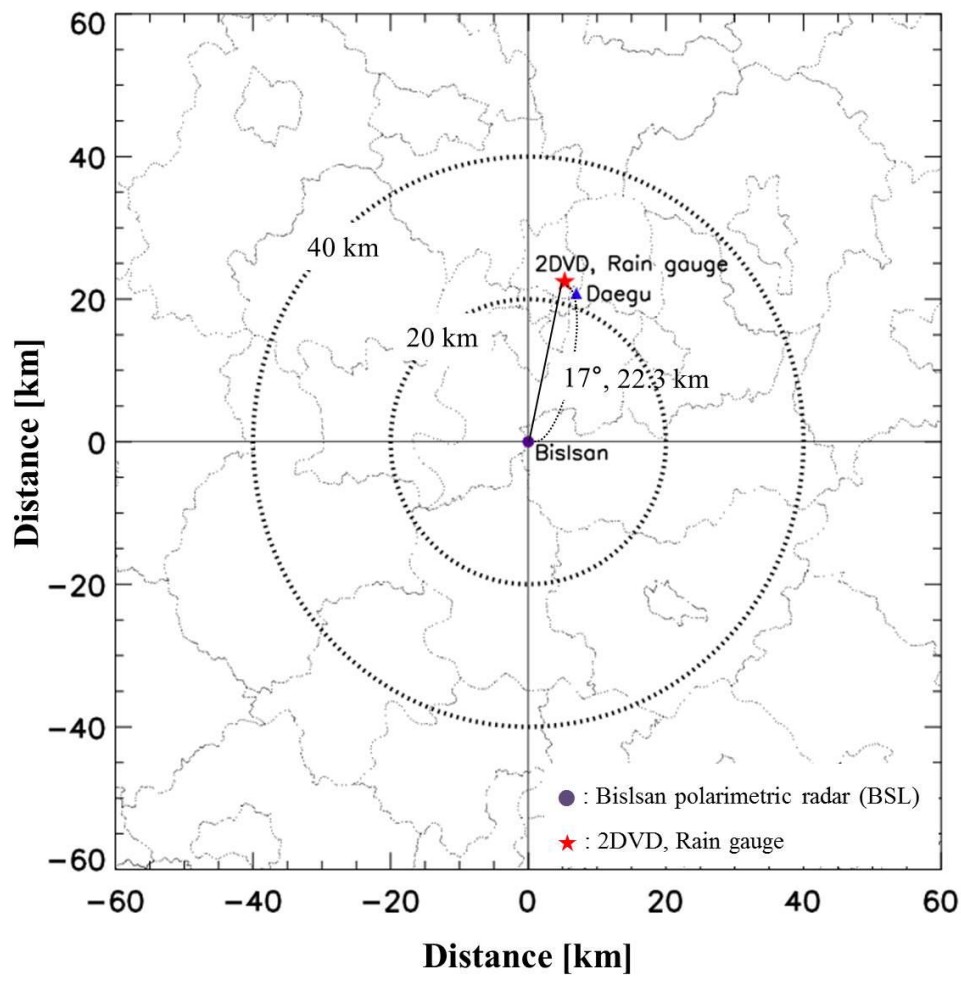

2      Figure 1. The location of the Bislsan polarimetric radar and the 2DVD with rain gauge site.





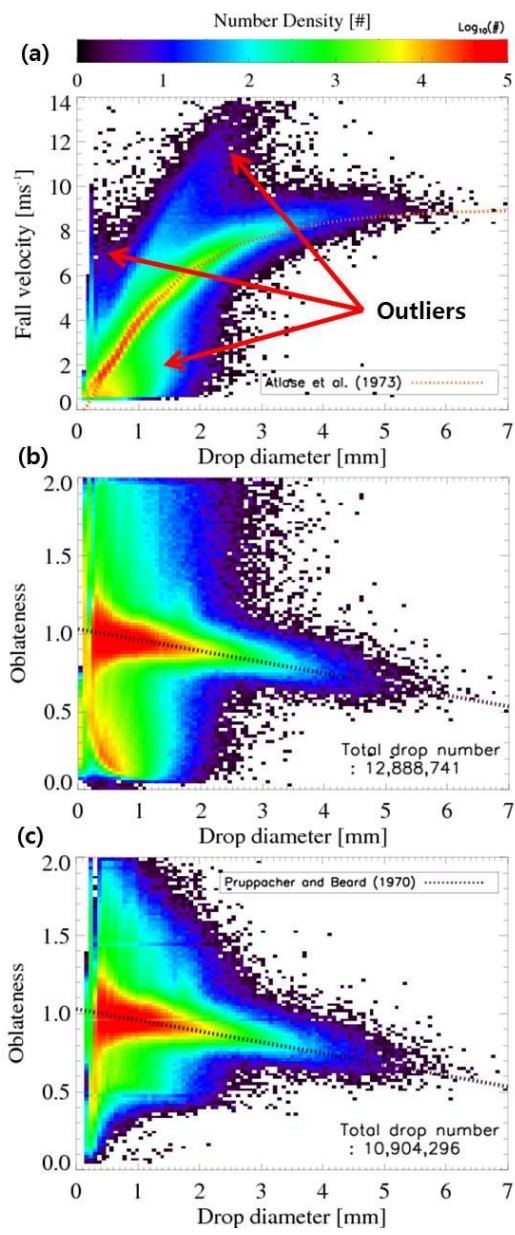

Figure 2. Distribution of fall velocity and oblateness according to drop diameter. (a) Velocity-
based filter for the drop measurements. The color scale represents drop number density (log
scale). (b) Drop axis ratios for all measured drops. (c) Drop axis ratios after removing
mismatched drops.





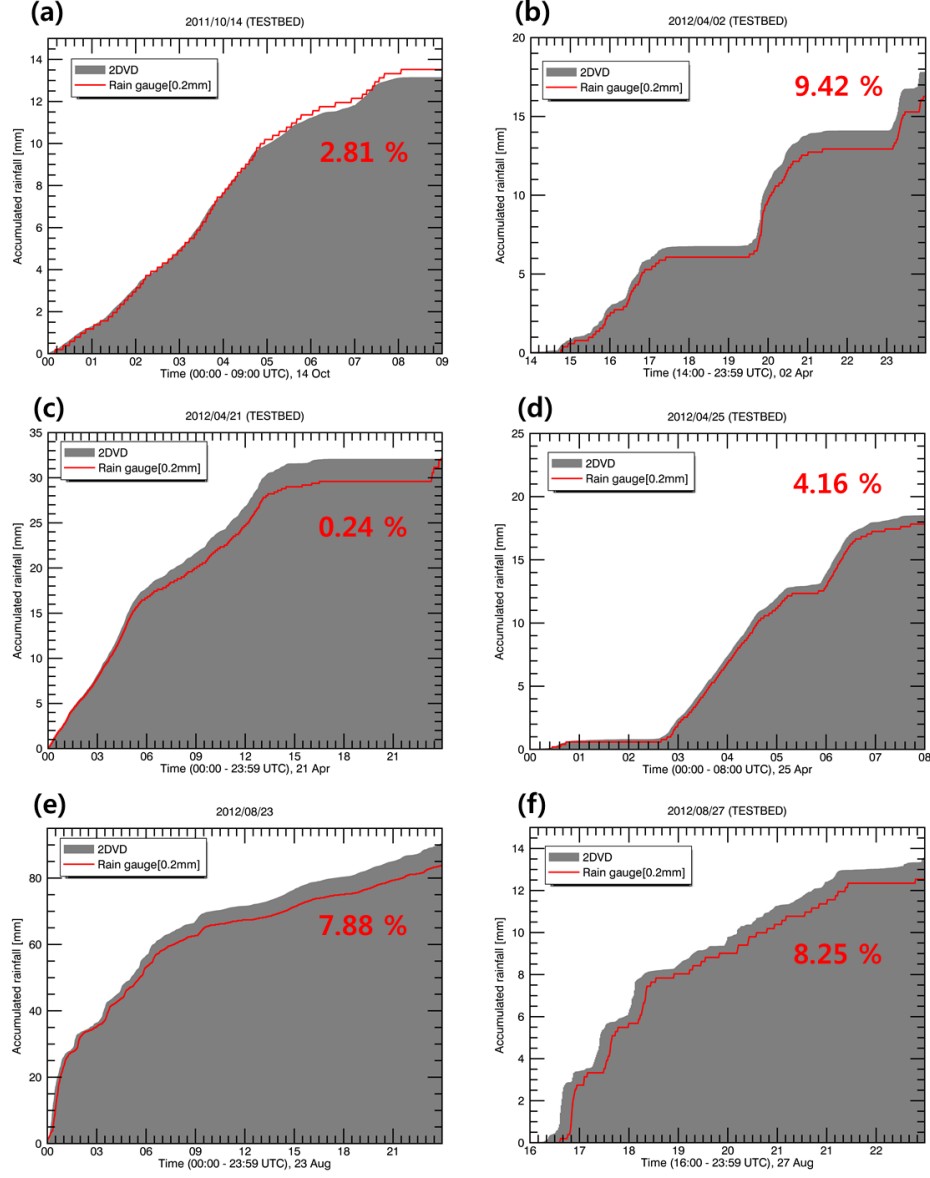

Figure 3. Time series of accumulated rainfall measured from the rain gauge and estimated
from the 2DVD: (a) 14 October 2011, (b) 2 April 2012, (c) 21 April 2012, (d) 25 April 2012.
(e) 23 August 2012, and (f) 27 August 2012.





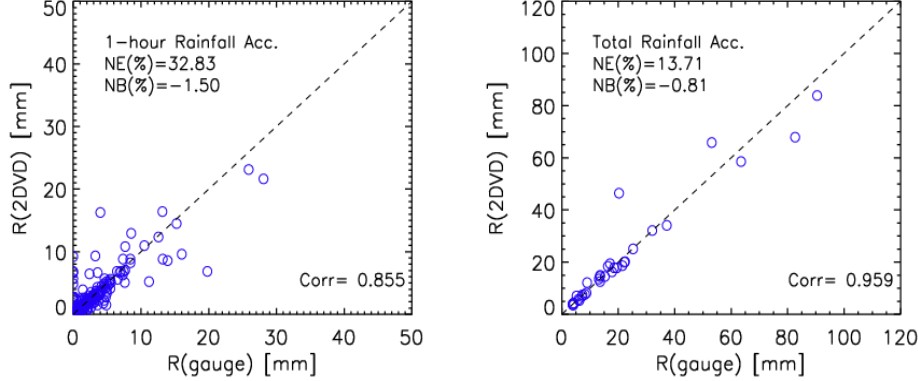

Figure 4. One-hour (left) and total accumulated rainfall (right) of 2DVD and rain gauge for
the 33 rainfall cases.

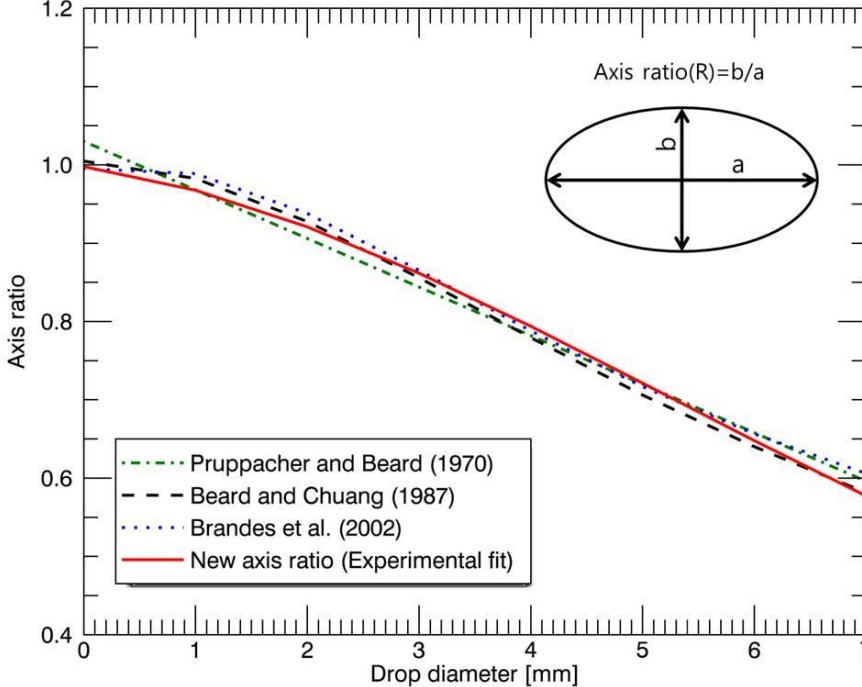

Figure 5. Different raindrop axis ratio relations for the oblate raindrop model. The upper right
subfigure illustrates the axis ratio of an oblate raindrop.





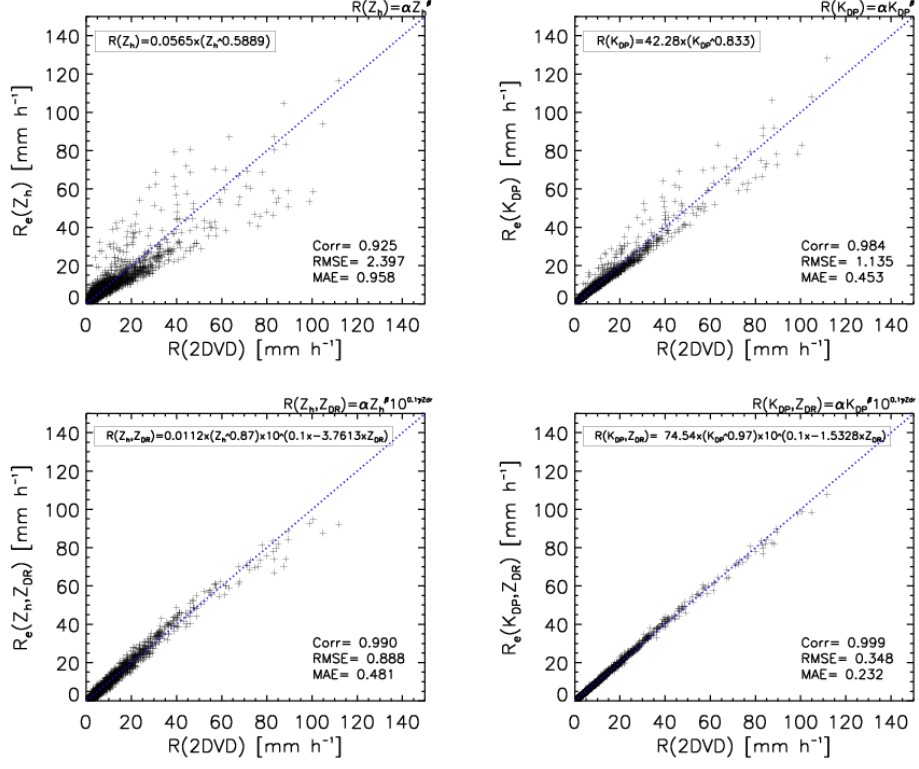

2    Figure 6. Scatterplot of R derived from observed DSDs of 17,618 min and $R_e$ estimated from

3    combinations of polarimetric parameters. $R_e$ is then obtained from the same dataset.



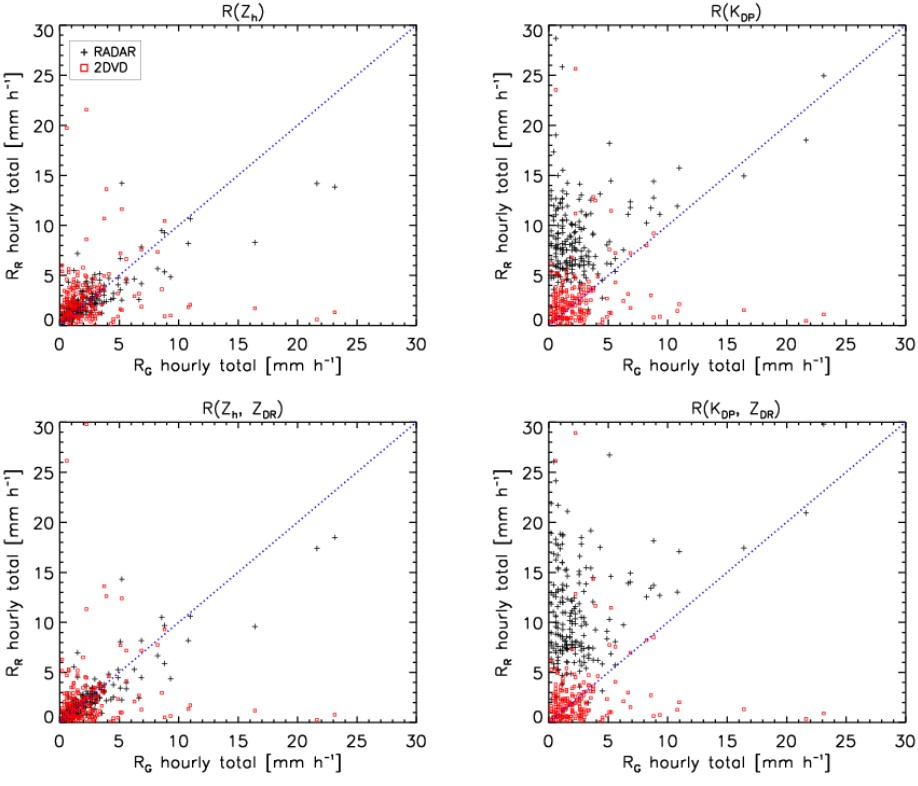

Figure 7. Scatter plot of one-hour rain rate from rain gauge ($R_G$) and BSL S-Band radar (or
2DVD) based on Eq. (4) for 18 rainfall cases: The pluses represents one-hour gauge rain rate
versus radar hourly rain rate from polarimetric rainfall algorithms, and squares indicate gauge
and 2DVD rain rate by different polarimetric rainfall algorithms.





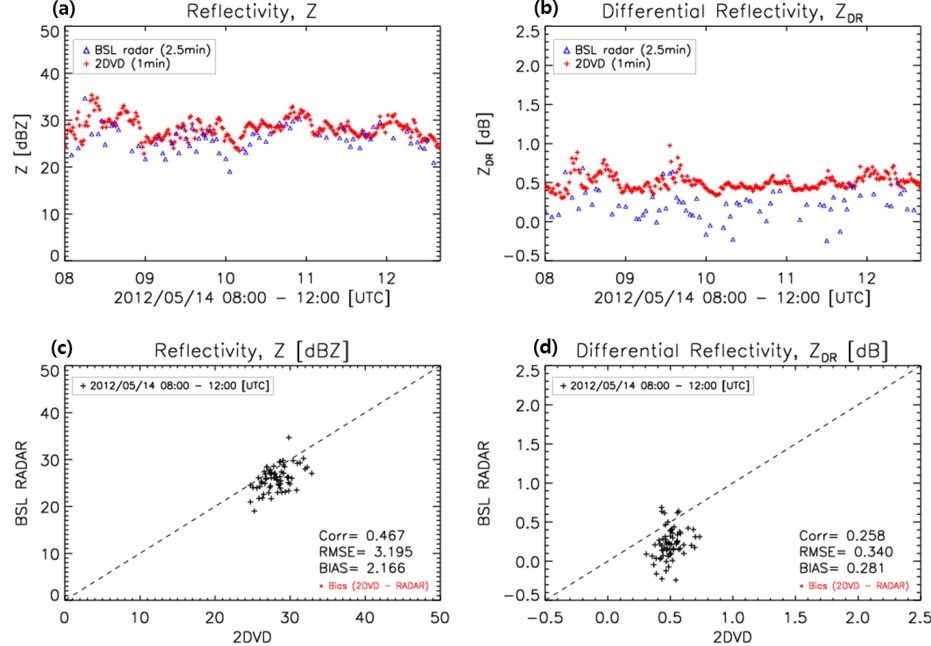

Figure 8. Time series of the (a) reflectivity, and (b) differential reflectivity by 2DVD and BSL
S-Band radar: Scatter Plots of the 2DVD estimation and radar measurement for the (c)
reflectivity and (d) differential reflectivity. Comparison statistics including correlation
coefficient (Corr), RMSE, and bias are also presented (14 May 2012).





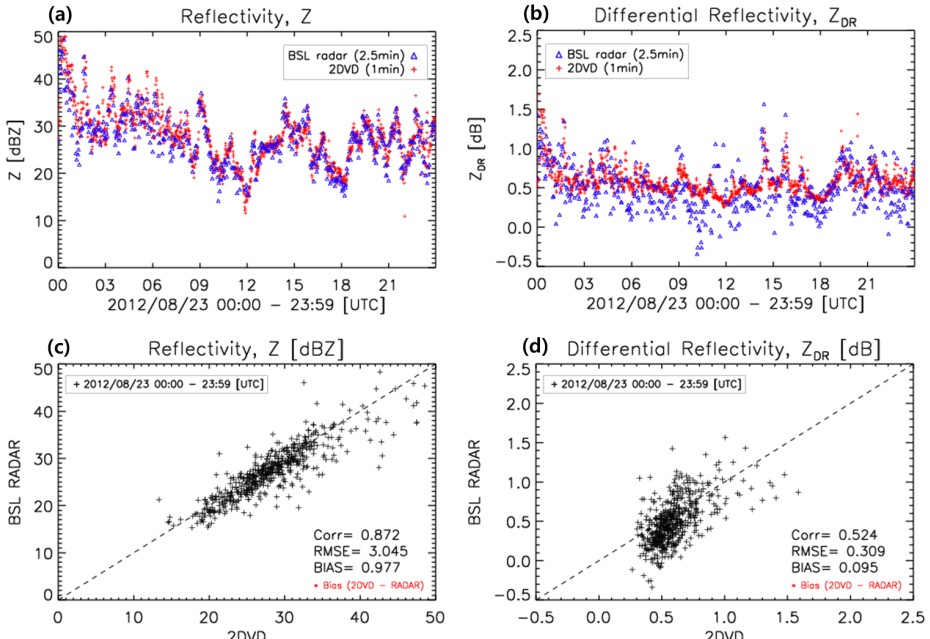

2    Figure 9. Same as Fig. 8, except that the data is for 23 August 2012.





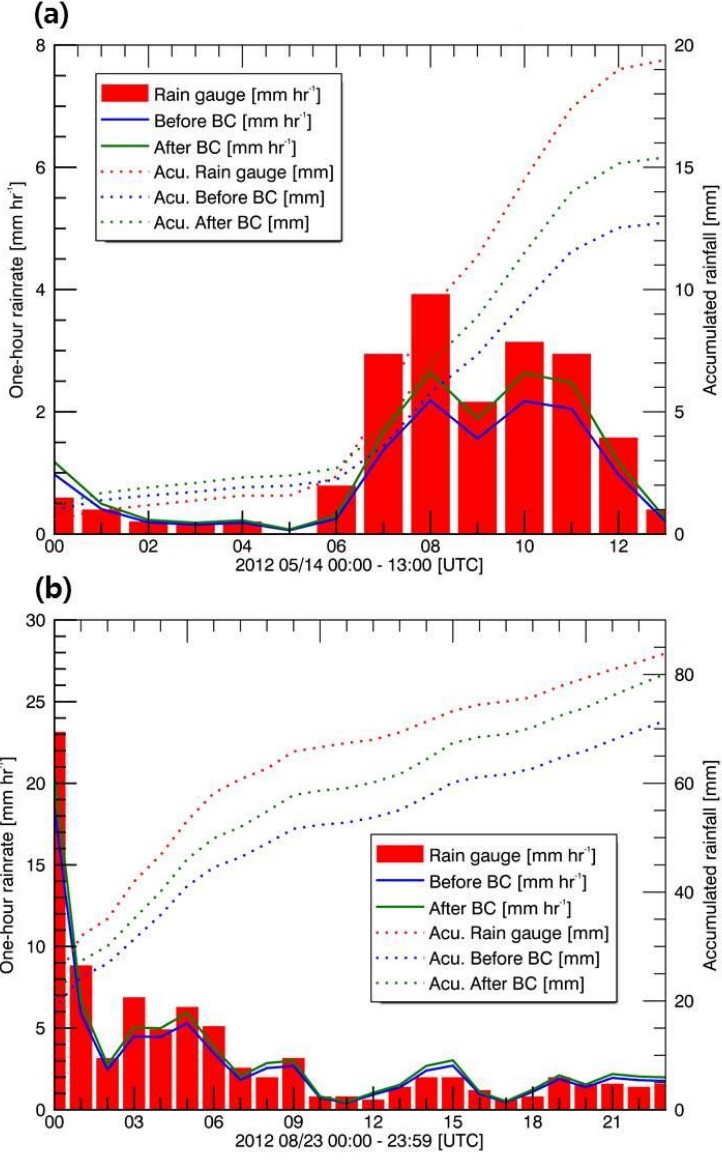

Figure 10. Comparison of the one-hour rain rate (left ordinate) and accumulated rainfall (right
ordinate) obtained by BSL S-Band radar and rain gauge: The R(Z$_H$, Z$_{DR}$) algorithm based on
Eq. (4) was used for rainfall estimation for (a) 14 May 2012 and (b) 23 August 2012. BC
represents Bias Correction.