# Peer review of "Manuscript under review for journal Atmos. Meas. Tech."

_Atmospheric Measurement Techniques, 2016_

## Referee Comment (RC1) · Anonymous Referee #1 · 10 Feb 2016

**Dual-polarization radar rainfall estimation in Korea according to raindrop shapes using a 2D Video Disdrometer**

by H.-L. Kim, M.-K. Suk, H.-S. Park, G.-W. Lee and J.-S. Ko

**Summary:**
The accuracy of different dual-polarization rainfall retrieval algorithms is quantified using data from the Bislsan S-band weather radar and independent measurements from a nearby video-disdrometer and co-located rain gauge. A new raindrop axis-ratio model that better reflects the characteristics of rainfall on the Korean Peninsula is derived (using the 2DVD data) and its influence on the rainfall retrievals is tested. The results show that drop-axis ratios have a small yet non-negligible impact on radar-retrieval algorithms. The most important factors affecting the accuracy seem to be (1) the calibration of the weather radar and (2) the comparison of the gauge and 2DVD data with a relatively large radar volume.

**General Comments:**
The results are interesting but not exactly new or groundbreaking. In particular, it is unclear whether the new axis-ratio proposed by the authors is actually necessary or not (given the strong uncertainties and large errors affecting radar measurements). There is nothing wrong with the approach. But the conclusions are relatively weak and the central message of the paper needs to be stated more clearly.

**Recommendation: Minor revision**

**Specific Comments:**
- Page 3, ll.29-30: *"Thereafter, improvement of quantitative rainfall estimation was investigated by applying derived calibration bias."* This sentence is not clear. Please reformulate.
- Page 4, Section 2.1 (Disdrometer): It might be worth mentioning here that the 2DVD is considered one of the best and most reliable disdrometers on the market today.
- Page 5, Section 2.3 (Rain gauge): Please provide the brand/make and model number of the tipping bucket and specify if the data were quality controlled or not.
- Page 6, ll.10-11: Please explain why the rain rates from the 2DVD data are computed using the Brandes et al. (2002) velocity model while the older Atlas et al. (1973) velocity model is used to filter the 2DVD data.
- Page 6, ll.24-25: *"Therefore, the 2DVD data within 20%  error were used in this study."* I'm not sure to fully understand what you mean by this. Are those 20% with respect to hourly accumulations or on an event basis?
- Page 7, ll.22-23: Here, it might be worth to say what you actually mean by "drop diameter" in this context. I assume you are referring to the diameter of a sphere with equal volume.
- Page 7, ll.27-28: Please provide at least one good reference for the T-matrix method.
- Page 9, l.2: *"The polarimetric radar contains systematic bias of the radar itself."* Not sure what you mean by this. Please reformulate.
- Page 10, ll.5-6: *"This means that raindrops in South Korea are more oblate than the others.".* This statement needs to be reformulated. There are many possible explanations for this and it would be premature to conclude that raindrops in South Korea are more oblate than in other places. The differences in axis-ratio might also be the result of instrumental effects, drop filtering and event selection. Please reformulate.
- Page 10, l.21: The correlation value of 0.10 mentioned in the text seems to be incorrect.
- Page 11, Eq.(12) and (13): There is no need to repeat the definition of the MAE and RMSE here.
- Page 12, ll.6-7: *"In addition, the radar rainfall estimations from R(Kdp) and R(Kdp,Zdr) perform better than those of R(Zh,Zdr) for rain rates exceeding 5 mm/h"*. This is not obvious from the graph.

Please provide hard evidence to back up this statement (e.g., in the form of an additional table or RMSE values for R>5 mm/h).
- Page 12, ll.14-15: *"Therefore, rainfall characteristics should be reflected in polarimetric rainfall relations."*. This is too vague. Please reformulate.
- Page 20, Table 3: Please check if the low correlation values (0.10) are correct.

**Typos and English:** (this is not an exhaustive list)
- Page 1, ll.15-16: The shapes of raindrops play an important role in  polarimetric rainfall algorithms  based on differential reflectivity (Zdr) and specific differential phase (Kdp).
- Page 1, l.21: In this study, we  present a method …
- Page 1, l.23: First, a new axis ratio of raindrop relations  is developed …
- Page 1, ll.24-27: Second, polarimetric rainfall algorithms  are derived using different axis ratio relations, and estimated radar-point one-hour rain rate for the differences in polarimetric rainfall algorithms  are compared with the hourly rain rate measured by  a rain gauge.
- Page 1, ll.27-28: In addition, radar rainfall estimation  is investigated …
- Page 2, ll.1-2: The R(Kdp,Zdr) algorithm based on  the new axis relation …
- Page 2, ll.24-25: This is because the shape of raindrops is one of the most sensitive parameters for representing the scattering properties of rain
- Page 3, l.17: In addition to  disdrometers, there are …
- Page 3, ll.22-23: In this study, we developed a mean axis relation and polarimetric rainfall algorithms using 2-Dimensional Video Distrometer (2DVD) measurements …
- Page 3, l.24: The four raindrop shape assumptions
- Page 3, ll.26-27: accurate polarimetric rainfall retrieval algorithms
- Page 3, l.28: with simulated Zh and Zdr obtained from the 2DVD.
- Page 4, l.15-16: From these, one can calculate the DSD and all related quantities such as the rain rate, total drop number concentration and liquid water content.
- Page 4, l.22: The BSL S-Band radar  measures polarimetric variables  such as Zh, Zdr, Kdp and rhohv. The considered data  are taken from six elevation angles.
- Page 6, l.10 is the drop interval.
- Page 6, l.18: The 2DVD recorded 13.14 mm and the rain gauge …
- Page 6, l.27: over the  Korean Peninsula.
- Page 7, l.1: compared to  in situ rain gauges.
- Page 7, ll.6-8: The overall agreement  between the 2DVD and the rain gauge was good. The total accumulated rainfall recorded by the gauge was larger than that of the 2DVD by about 0.81%.
- Page 7, l.17: could reach  about 8 mm, …
- Page 8, l.17: there is a space missing between Table and 3.
- Page 8, ll.15-16: … are derived when the rain rate is …
- Page 10, l.15: where R is the rain rate from observed 1-min DSDs and Re is the rain rate from  various combinations of polarimetric parameters.
- Page 13, l.5: was improved by about 13.71%.
- Page 13, l.6: The recorded accumulated rainfall  was
- Page 13, l.8: was improved by about 10.25%.

---

## Referee Comment (RC2) · Anonymous Referee #2 · 15 Feb 2016

**REVIEW REPORT**

Review of amt-2016-14
By H. -L. Kim, M.-K. Suk, H. -S. Park, G. -W. Lee, and J. -S. Ko
Manuscript Title – Dual-polarization radar rainfall estimation in Korea according to raindrop shapes using a 2D Video Disdrometer

**GENERAL REMARK**

In this study one year of 2D Video Disdrometer (2DVD) data collected in Korea has been used to establish: *i)* a new axis ration versus diameter relation and *ii)* weather radar algorithms for rainfall retrieval optimized for that area. They verified the adequacy of the new axis-ratio comparing the rainfall rate estimated through different radar rainfall algorithms and assuming different axis ratio relations. Furthermore the Authors compared the rainfall retrieved from a Bislsan S-Band dual-polarization radar and from 2DVD through the obtained algorithms with the one measured by a rain gauge nearby. Finally in order to improve the radar estimates, they used the 2DVD data to adjust the horizontal reflectivity and differential reflectivity measured by S-band radar. I recommend the publication of the paper on Atmospheric Measurement Technique (AMT) after the revision. Shown below some specific comments and questions.

**SPECIFIC REMARKS**

1. Pag. 6, line 10: Why the Authors used the fall velocity-diameter relation of Brandes et al. (2002) to compute the rain rate and the one of Atlas et al. (1973) to filter out the outliers? Defend this choice or use the same relation.
2. Pag. 6, line 12: How did the authors define an event? please provide some information regarding the criterion used.
3. Pag. 6, line 26: How many events have been discarded by the quality control process?
4. Pag. 7, lines 1-3: In the literature there are several different methods to distinguish between stratiform and convective rain. Can the Authors provide some information regarding the criterion used?
5. Pag. 7, Line 19: Can you provide some information regarding the choice of a third order polynomial? Did the Authors try also other relations (such as linear or a fourth order polynomial)? Can the Authors provide some information regarding the performance/goodness of the fitting (such as $R^2$)
6. Pag. 8, line 13: Can the Authors defend the choice of using $D_{max} = 7$ mm in the T-matrix simulation instead of using the maximum drop diameter measured by 2DVD in each DSD?
7. Table 3: Please note that some correlation coefficients of Table 3 are very poor (0.01) please check this values.
8. Pag. 10, lines 5-6: "This means that raindrops in South Korea are more oblate than the others". Please rewrite this sentence, it is "too strong". I think that there are not enough information to justify it.
9. Pag 10, line 21: "Corr = 0.01". Please check this value.

10. Table 3: In terms of MAE, RMSE and corr., it seems that the use of different axis-ratio does not have a huge influence, in particular the use of "new axis ratio" (that is optimized for the area) does not highly improve the three statistics. Can the Authors defend this issue?

11. Pag 11, Line 11: Why the statistical validation has been performed only on 18 events. How do you chose them?

12. Table 4: As for Table3, also in this case the use of the "new axis ratio" does not produce an high improvement, in particular with respect to the relation of Pruppacher and Beard (1970). Please comment this issue.

13. Pag. 12 Line 1-2: the sentence is not clear to me. Which are the "DSD results"? How the Authors can say that, based on the results provided, "$K_{dp}$ is less sensitive to DSD variation and uncertainties in raindrop shape"? Please explain in more detail.

14. Pag. 12, line 7: I suggest to show some plots regarding the performance of the algorithms for $R > 5$ mm h$^{-1}$

15. Pag. 12 line 9-10: This is known in the literature (such as Vulpiani et al. 2015 among others), please provide some reference.

16. Pag. 12, line 11-15: "the polarimetric rainfall relations based on the new axis-ratio relation also were better than the others". Please explain how you can say that they are better. I think that this paragraph is about a crucial issue however it is not clear to me. I suggest to rewrite in order to explain the improvement of the new axis ratio in the rainfall retrieval.

17. Section 4.2.3: I think that the world "calibration" is misleading, the procedure explained in the paragraph is not really a calibration procedure, in my opinion it is more an adjustment of radar variables based on a ground truth (2DVD data). Pleas reformulate it. Moreover, the Authors did non compute the "calibration bias" for $K_{dp}$, why?

18. Pag. 12, line 18: why do you use only 8 events for this analysis? Can you use all the 18 events selected for the validation? I noticed that in the 8 selected events there aren't convective events. Why?

19. Pag. 12, Line 19-21: This is just a suggestion. Instead of doing two scatterplots for each event (namely Figure 8 *c* and *d* for one event and Figure 9 *c* and *d* for another one) the Authors can plot the results for all the events in two graphs (one for $Z_h$ and one for $Z_{dr}$).

**EDITORIAL REMARKS**

1. Pag. 1, Line 16-17: "The shapes of a raindrops have a direct impact on rainfall estimates". Although the sentence is correct in my opinion it is necessary to add that also the number, the dimension (diameter) and fall velocity of the raindrops play a central role.

2. Pag. 1, Line 20: I suggest to write "raindrop size distribution" instead of "raindrop shapes".

3. Pag. 2, Line 2; what does it means "DSD statistics"? These worlds are used several times in the manuscript, but it is not clear the exact meaning. Please clarify it.

4. Pag. 2, line 24-25: "This is because the shape of raindrops is one of the most sensitive parameters for representing the DSD properties of the rain". Please consider to reformulate this sentence or to put one or more reference. There are other important parameters (e.g. diameter, number of drops ecc) that should be mentioned.

5. Pag. 3, lines 7-10: Please check this sentence, single polarimetric radar does not provide $Z_{dr}$

6. Pag. 3, Line 24: I think that "four" should be substitute to "three"
7. Pag. 6, Line 3-4: This sentence should be moved to pag. 5 Line 25 after "….Atlas et al. 1973)". Furthermore it is not clear the meaning of "the normal distribution". Please reformulate it.
8. Pag. 7, Line 13-14: similar to my previous comments, I think that the sentence "this relation is one of the most sensitive parameters for representing the rainfall properties" is "too strong". Can you provide some references or reformulate it?
9. Table 5: check the format of the date.

REFERENCE

Vulpiani, G., L. Baldini, and N. Roberto. "Characterization of Mediterranean hail-bearing storms using an operational polarimetric X-band radar." *Atmospheric Measurement Techniques* 8.11 (2015): 4681-4698.

---

## Referee Comment (RC3) · Anonymous Referee #3 · 23 Feb 2016

Review report of

By H. -L. Kim, M.-K. Suk, H. -S. Park, G. -W. Lee, and J. -S. Ko: Dual-polarization radar rainfall estimation in Korea according to raindrop shapes using a 2D Video Disdrometer (amt-2016-14)

**General comments**

The paper by Kim et al. aims at demonstrating the improvement dual polarization radar quantitative precipitation estimation through: a) defining rainfall algorithms using DSD from 1 year of 2D video disdrometer measurements and defining an optimal shape-size relation based on the same 2DVD measurements; b) using 2DVD measurements to compensate bias in Zh and Zdr. The performance of radar rainfall algorithms is tested by comparing 2DVD rainfall estimates to rainfall obtained from intrinsic polarimetric measurements obtained by applying T-matrix simulation to 2DVD-estimated DSD and using measurements collected by the S-band Bislan radar.

Radar rainfall algorithms are derived and tested using the same DSD dataset and therefore should be optimal. Differences in performance should point out the benefit of using the new shape-size relation. The improvement is not very evident and seems overwhelmed by error measurements, especially those related to the estimation of $K_{dp}$, which appear very high. Results are summarized by a couple of tables, but the advantage of using the new shape-size relation is not evident at all. Table 3 shows that only in the case of the $R(K_{DP}, Z_{DR})$ algorithm there is an improvement. Moreover, using actual radar measurements, improvement is only appreciable for $R(Z_H, Z_{DR})$. A step that is missing is a simulation including effects of biases and fluctuations. However, several papers are available on

A calibration of radar using disdrometer measurement is obtained using disdrometer-derived measurements of $Z_h$ and $Z_{dr}$ and corresponding radar measurements While there are some doubts about the meaningfulness of this calibration, especially as $Z_{dr}$ is concerned, calibration should be applied before evaluating the performance of rainfall algorithms.

Summarizing, the paper does not present any significant improvements with respect to the state of the art. More important is not clear what is the actual aim of the paper. Although understandable as a whole, at least for evaluation purposes, the paper presents several unclear sentences and some typos.

**Specific comments**

Pag. 1, lines 19-21: It is not clear the link of the sentence starting with "therefore" with the previous one

Pag. 2, lines 16-18: Potential and actual advantages of polarimetry applied to radar systems have been analyzed in the literature. However, this sentence is not very precise.

Pag. 2, lines 20-21: Dual polarization rainfall algorithms use $K_{dp}$ or combinations of $Z_h$, $Z_{dr}$, and $K_{dp}$.

Pag. 2, lines 24-25: This is wrong: maybe "allowing correct interpretation of polarimetric measurements in rain" or "an important feature of rain microphysics".
Pag. 2, lines 26-28: Authors should cite also more recent work on the drop-shape topic.
Pag. 3, lines 1-3: "Thus,.." Again, I do not see how this conclusion follows from the previous sentence.
Pag. 3, lines 14-17: Cited studies by Goddard et al. aimed at demonstrating the inadequacy of the Pruppacher and Beard shape-size relation.
Pag. 3, line 19: Formulation of self-consistency by Gorgucci et al. 1992 was based on $Z_h$, $Z_{dr}$ and $K_{dp}$ and not on Zh-$K_{dp}$.
Pag. 3, line 21: A recent paper by Chandrasekar et al. (2015) is more appropriate and update than Atlas (2002).

Pag. 3, line 24: Replace "four" with "three"

Pag. 4, line 20: Please add the height ASL of the radar

Pag. 5, lines 1-2: A 0-deg elevation allows small distances between measurements aloft and ground measurements. For most installations measurements collected at such small elevation angles are prone to effects of nearby obstacles. Please demonstrate that for the Bislan radar, this elevation does not implies beam blocking/ground clutter effects or that they are negligible.

Pag. 5, lines 2-5: Please explain how averaging of $PHI_{dp}$ is obtained.

Pag. 5, line 21: What does it mean "beyond the normal distribution" ?

Pag. 6, line 10-11: Why do Authors use the velocity-size relation by Brandes et al. (2002) and Atlas et al. (1973) to filter 2DVD measurements? Note that, starting from 2DVD counts, such relation is not necessary for computing R.

Pag. 7. Section "Raindrop axis ratio": There are a number of questions about the fitting (4). First, what is the accuracy of the fitting. Second, is this fitting more appropriate for certain events (i.e. is there an event-by-event variation?).

Pag. 7. Section "Disdrometer-rainfall algorithms": I think it is more appropriate to $Z_h$, $Z_{dr}$ $K_{dp}$ "variables" or "measurements" instead of "parameters"

Pag. 8, line 12. Likely values of mean and standard deviation are switched.

Pag. 9, section 3.4. What is the point of using light rain ? $Z_{dr}$ near zero ? Please explain. What is the accuracy expected with this calibration ?

Pag. 10, "Variability of DSD in rainfall estimation" I would like to see also some relative performance factors, such as the ratio of RMSE and average value of R.

Pag. 10, line 21: Corr = 0.1 ?????

Pag. 11. Line 3. "A summary of ....." The beaviour of the different algorithms with intrinsic dual-pol measurements is what is expected (eg. Bringi and Chandrasekar, 2001). What is strange is that only R($K_{DP}$, $Z_{DR}$) takes advantage from the new shape-size relation.

Pag. 11. Line 21-22. Now, it is R($Z_H$, $Z_{DR}$) the best algorithm and is the only one that take advantage from new shape-size relation. This is also not surprisingly. A simple exercise consisting in adding a properly modelled error to intrinsic measurements would reveal how algorithms are sensitive to random measurement fluctuation and/or calibration biases (see again Bringi and Chandrasekar, 2001). The bad performance of R($K_{DP}$, $Z_{DR}$) can be ascribed to the an unappropriate $K_{DP}$ estimation (see the increase in the error of radar R($K_{DP}$). From Figure 7 compared with Figure 6, I would expect worst MAE and RMSE values than those in Table 4. Finally, it is not clear to me whether $Z_h$ and/or $Z_{dr}$ bias correction was applied or not here.

Pag. 12, section: "Correction of calibration bias": What is the accuracy of this calibration ? Can the event-to-event variability of the bias be related to variation of radar performance ? Figure 9 shows clearly that the estimation of $Z_{dr}$ bias is extremely poor.

**References**

E. Gorgucci, G. Scarchilli, and V. Chandrasekar, Calibration of radars using polarimetric techniques, IEEE Trans. Geosci. Remote Sens., vol. 30, no. 5, pp. 853–858, Sep. 1992

Chandrasekar V., Baldini, L., Bharadwaj N., Smith, P.L., Calibration Procedures for Global Precipitation-Measurement Ground-Validation Radars, TheRadio Science Bulletin No 355 (December 2015), pp. 45-73 [http://www.ursi.org/files/RSBissues/RSB_355_2015_12Corrected.pdf]

V. N. Bringi and V. Chandrasekar, Polarimetric Doppler Weather Radar: Principles and Applications . Cambridge, U.K.: Cambridge Univ. Press, 2001, p. 648

---

## Referee Comment (RC4) · Anonymous Referee #4 · 28 Feb 2016

**1 General comments:**

This paper presents work to determine and test appropriate dual-polarisation radar rainfall-rate algorithms with for an S-band radar, based on radar, 2DVD and rain gauge measurements. Adding additional drop-size/axis-ratio relationships to the published literature is always potentially useful, as is the results of practical attempts to compare different possible dual-polarization rainfall-rate estimation algorithms. Given that many national meteorological agencies are in the process of implementing dual-polarization rainfall-rate estimation algorithms, the publication of what one particular organisation is doing is likely to be very helpful to others.

[Figure]

However, I have some serious reservations about the core of this paper: the need for a specific drop-size/axis-ratio relationship for Korea. I agree that the raindrop axis ratio is one of the parameters to which dual-pol radar measurements are particularly sensitive. However, the need to have different drop-size/axis-ratio relationships for different regions still seems to be an open research question. You appear to conflate the well-documented regional variation in DSDs with variation in drop-size/axis-ratio relationships, and assume without comment or explanation that there is indeed significant regional variation in drop-size/axis-ratio relationships.

The model given in the Beard and Chuang 1987 paper that you cite essentially gives that drop shape varies with temperature and pressure. Average temperatures and pressure do of course vary from region to region, and would therefore give rise to some variations in drop shape around the world. Some (unpublished) calculations of my own, based on an implementation of the Beard and Chuang 1987 model suggest that a pressure change from 97325 to 105325 Pa results in a 0.006% change in axis ratio for a 6 mm volume-equivalent-sphere diameter drop, which is quite insignificant and can be neglected. For a temperature change from 0 to 20°C I find a percentage change of about 1.1%, which is potentially a little more significant. However Thurai and Bringi 2005 report a standard deviation in their measured axis ratio for 6-6.5 mm diameter drops of around 11% of the axis ratio. I therefore suggest that temperature variation in drop shape is likely small compared to the natural variation in drop shape at that temperature, and so is not likely to be a major factor.

The only works that I have come across in the literature that specifically address regional variation of drop-size/axis-ratio relationships are Marzuki et al. 2013 and Gorgucci et al. 2009. The former is concerned with 2DVD measurements in Sumatra, and does find a somewhat different drop-size/axis-ratio relationship than others in the literature, although they also express concerns about the siting of their 2DVD. They suggest that atmospheric conditions are unlikely to be a significant factor, but suggest that variations in the amount of raindrop collisions may be a possible cause. Gorgucci

et al. 2009 present a method for determining drop-size/axis-ratio relationships from radar measurements, and report similar relationships from Brazil and Italy, but a different relationship for Florida (albeit all in line with other, previously reported values in the literature), but I don't think their results could be taken as definitive. They also suggest drop collision and the resultant oscillations as a possible source of the variability.

Additionally, the work of Thurai and Bringi 2005 was carried out under very controlled conditions, with careful repeated calibrations of the 2DVD. I am concerned that a subset of a year's worth of data under less controlled conditions (especially for wind speed and calibration) is unlikely to be as accurate as Thurai and Bringi's work and that, whilst it is possible that a notably different drop-size/axis-ratio relationship exists in Korea, the measurements here may not be sufficiently good as to be able to prove it. I would also point out that in Thurai and Bringi 2005 and Thurai et al. 2007 a significant amount of effort in to explaining their calibration and interpretation of the 2DVD measurements made, aspects that you cover in a lot less detail.

At the moment, whilst you see clear benefits to your new relationship when comparing (what I take to be) purely 2DVD derived parameters (table 3), you see very little improvement in the actual radar measured values (table 4), which suggests to me that the different axis ratio is either not different enough to be significant, or that the measured difference is at least partly an artefact of the 2DVD data and not fully representative of the real rain. The very small improvement that you do see may be down to deriving the relationship for these rain events at the disdrometer, and then carrying out the evaluation on the same dataset.

I think there are two possible options here. The first is to provide a robust analysis demonstrating that you believe your measurement values to be accurate to a level that proves that the axis ratios that you have determined are accurate to a level that allows significant differentiation from the other axes ratios in the literature (of course, assuming that is the case). This would also require a significant beefing up of your explanation of how you determine axes ratios from the 2DVD data, or at least the

uncertainties associated with that. You would also need to provide more motivation for the regional variability of the relationship - references, possible mechanisms for the variability, etc. A positive demonstration of significant regional variation in drop-size/axis-ratio relationships would in my opinion be a very interesting result.

Alternatively, you could treat the measured axis ratio relationship more as a slight variation of of those already in the literature (it is very close). You could then focus the paper more on the dual-polarisation algorithms being developed and deployed in Korea - information on what a national meteorological organisation is implementing is in my opinion interesting and worthy of publication in its own right.

**2 Specific comments:**

Page 2, line 29: I don't agree that "[t]he raindrop shape is defined by the shape-size relationship of a raindrop". The shape is rather defined by interactions between the drop and the atmosphere (and other drops). However, the average shape of a raindrop can be inferred from its size.

Section 2.1: Additional information about the siting of the 2DVD should be given here: things like closeness to buildings could be significant.

Page 5, line 1: You say that you use the 0.0° elevation "to avoid effects from beam blocking and ground echoes". Surely that would be further reduced by using the 1.6° elevation data? You should say why you opted for 0.0° rather than 1.6° (to have measurements as close as possible in space to the 2DVD?).

Figure 3: You seem to be showing some of the best available cases. Better might be to show a representative sample, including some of the best and some of the worst.

Section 3.2: You should give more information on the dataset you were working with here. What size of diameter bins were you using, what was the minimum number of

drops in a bin included in the final fitting, what were the axis ratio standard deviations, etc.?

Page 8, line 12: It may just be a typo, but I think you should be using a mean canting angle of $0°$ and a standard deviation of $7°$, not the other way round.

Section 3.4: It is not quite clear here whether you are applying these calibration corrections only in table 5, or also in table 4. If in table 4, then assuming the new axis ratio calibration values in all cases would perhaps unfairly disadvantage the other axis ratios considered. Additionally, you should explain why you consider only light rain events for the calibration.

Section 4.2.1: I think that here Re is being determined by applying the various algorithms to radar polarimetric variables derived from the 2DVD data. Is that correct? Either way, you should make this clearer.

Table 3: For the Kdp, Zdr algorithm, I suspect the 0.1 correlation values are wrong (compare with figure 6).

Section 4.2.2: Why do you carry out this validation only on 18 of the rain events and not all 33?

**3 Technical corrections:**

Page 1, line 26: I think you mean different not differences here.

Page 6, line 10: I think you mean that the drop fall velocity formula used was that derived by Brandes et al.

Page 7, line 1: In situ rather than "in suit".

**4  References:**

Thurai, M. et al.: "Drop Shapes, Model Comparisons, and Calculations of Polarimetric Radar Parameters in Rain", J. Atmos. Oceanic Technol. 24, 1019-1032, June 2007

Gorgucci, E., Chandrasekar, V. and Baldini, L.: "Can a unique model describe the rain-drop shape-size relation?  A clue from polarimetric radar measurements?", J. Atmos and Oceanic Technol. 26, 1829-1842, September 2009

Marzuki, Randeu, W.L., Kozu, T., Shimomai, T., Hashiguchi, H. and Schönhuber, M.: "Raindrop axis ratios, fall velocities and size distribution over Sumatra from 2D-Video Disdrometer measurement", Atmos. Res. 119, 23-37, January 2013

---

## Referee Comment (RC5) · Anonymous Referee #5 · 7 Mar 2016

General comments

The submitted paper talks about the development of a new axis-ratio relation with the 2DVD, wich is compared with other axis-ratio relations already known in the literature. The Authors applied also polarimetric rainfall algorithms to calculate the rain by utilizing as inputs polarimetric parameters, which are both measured by the radar and simulated by the disdrometer, according to the considered axis-ratio relations. Both radar and disdrometer rainfall estimates are then compared with rain gauges measurements. Another goal of the present work is to corrected horizontal reflectivity and differential reflectivity biases affecting radar estimates by comparing radar and 2DVD disdrometer polarimetric parameters. Gauge data are considered as truth. The topics of the paper

are ones of the most discussed in radar meteorology. Several past works, such as those cited by the Authors, have already dealt with the same problems giving valuable contributions in the assessment of radar estimates (see an additional reference list below for example). It's my opinion that the submitted work surely gives a contribution to the aforementioned topics. In fact, the element of originality is constituted by the fact the axis ratio relation proposed by the Authors can adapt to the particular weather conditions of the place in which it was developed, improving locally the radar estimates. So it's my opinion that some parts of the paper should be rewritten because poorly written (see specific comments and technical corrections). I also suggest to the Authors to submit the article after a revision concerning the English language. In conclusion, I believe that the manuscript it worth for publication after some major revisions.

Additional references

Lombardo, F., Napolitano, F., Russo, F., Scialanga, G., Baldini, L., and Gorgucci, E.: Rainfall estimation and ground clutter rejection with dual polarization weather radar, Adv. Geosci., 7, 127– 130, doi:10.5194/adgeo-7-127-2006, 2006b.

Sebastianelli, S., Russo, F., Napolitano, F., & Baldini, L. (2013). On precipitation measurements collected by a weather radar and a rain gauge network. Natural Hazards and Earth System Science, 13(3), 605-623, doi:10.5194/nhess-13-605-2013.

Zawadzki, I.: Factors affecting the precision of radar measurements of rain, in: Proceeding of the 22d Conf. Radar Meteorology, Zurich, Switzerland, 10–13 September 1984, Amer. Meteor. Soc., 251–256, 1984.

Villarini, G. and Krajewski,W. F.: Review of the different sources of uncertainty in single polarization radar-based estimates of rainfall, Surv. Geophys., 31, 107–129, 2010.

S. Spina, S. Sebastianelli, E. Ridolfi, F. Russo, L. Baldini, and L. Alfonso: Data selection to assess bias in rainfall radar estimates: An entropy-based method, AIP Conference Proceedings 1558, 1665 (2013); doi: 10.1063/1.4825849.

Zhang, J. and Qi, Y.: A real-time algorithm for the correction of brightband effects in radar-derived QPE, J. Hydrometeorol., 11, 1157–1171, 2010.

Zhang, J., Langston, C., and Howard, K.: Brightband identification based on vertical profiles of reflectivity from the WSR-88D, J. Atmos. Ocean. Tech., 25, 1859–1872, 2008.

Gorgucci, E., Scarchilli, G., and Chandrasekar V.: Calibration of radars using polarimetric techniques, IEEE Trans. Geosci. Remote Sens., 30, 853-858, 1992.

Specific comments and technical corrections

P1, title: I would suggest: "Dual-polarization radar rainfall estimation in Korea according to raindrop shapes obtained by a 2D Video Disdrometer".

P1, line 23: what did you mean with "a new axis ratio of raindrop relations"? It's a relation between the raindrop axis ratio and the raindrop diameter? Please specify. It's my opinion that you should better introduce this relation both in the abstract and in the introduction.

P1, lines 24-25: if you say: "polarimetric rainfall algorithms were derived using different axis ratio relations", it seems that the polarimetric algorithm changes depending on the axis ratio relation you consider. But if we observe table 3 you applied each polarimetric rainfall relation with the same 4 axis ratio relations. Instead, the polarimetric parameter depend on the shape of raindrops you assumed, as it's better specify at page 7 lines 28-29. So you should rephrase.

P1, lines 25-26: what did you mean saying "radar-point one hour rain rate"? A radar estimates rain in a pixel not in a point. A rain gauge measures rain in a point. Could you specify if the pixel in question includes within it the rain-gauge location? Furthermore with "one-hour rain rate" did you mean the average rain rate in a hour? It would be better if you put the unit. I would suggest to write "the radar-pixel hourly mean rain rate".

P1, lines 24-27: you wrote: "Second, polarimetric rainfall algorithms were derived using different axis ratio relations, and estimated radar-point one hour rain rate for the differences in polarimetric rainfall algorithms were compared with the hourly rain rate measured by gauge." The sentence is too long, please split it. Moreover, I suggest writing: "the estimated radar-pixel hourly mean rain rate obtained from the different polarimetric rainfall algorithms" instead of "estimated radar-point one hour rain rate for the differences in polarimetric rainfall algorithms"

P1, lines 27-28: I would change "in relation to calibration bias of reflectivity and differential reflectivity" into "to calibrate reflectivity and differential reflectivity biases".

P1, lines 28-29: the sentence is poorly written, I suggest to write: "For D < 1.5 mm and for D > 5.5 mm the shape of raindrops obtained by the new axis-ratio relation developer by the 2DVD is more oblate than the shapes obtained by the existing relations".

P2, lines 6-9: the sentence is poorly written, I would write "Zh and ZDR biases were calculated by comparing Zh and ZDR radar measurements with the same parameters simulated by the 2DVD. In order to produce more accurate rainfall estimation." (it is redundant to say that a bias is used to correct a bias). In addition it's my opinion that you should better specify what is the aim of your work. If I understood, you developed a new axis-ratio relation with the 2DVD. You developed also the polarimetric rainfall algorithms by using the same tool? Then, you applied polarimetric rainfall algorithms to calculate the rain by utilizing as inputs the polarimetric parameters, which are both measured by the radar and simulated by the disdrometer, according to 4 axis-ratio relations. You corrected also Zh and ZDR biases by comparing radar measurements of polarimetric parameter with the disdrometer simulations of the same parameters. Finally you validated the rainfall estimations obtained by both the radar and the disdrometer by a comparison with the rain gauges measurements considered as the ground truth. In many past works the rain gauge observations have been considered as the ground truth, for example in both Lombardo et al., 2006 and Sebastianelli et al., 2013 which I suggest the Authors to cite:

- Lombardo, F., Napolitano, F., Russo, F., Scialanga, G., Baldini, L., and Gorgucci, E.: Rainfall estimation and ground clutter rejection with dual polarization weather radar, Adv. Geosci., 7, 127– 130, doi:10.5194/adgeo-7-127-2006, 2006b.

- Sebastianelli, S., Russo, F., Napolitano, F., & Baldini, L. (2013). On precipitation measurements collected by a weather radar and a rain gauge network. Natural Hazards and Earth System Science, 13(3), 605-623, doi:10.5194/nhess-13-605-2013.

In particular, the radar estimations reliability is assessed before and after the calibration to test the effectiveness of the calibration process. Could you better clarify this aspect in the text by briefly describing the methodology followed in your work?

P2, lines 13-16: I would write: "In particular, a dual polarization radar can estimate rainfall more accurately than a single polarization radar by providing reflectivity (ZH), differential reflectivity (ZDR), differential phase ($\Phi$DP), specific differential phase (KDP), and cross-correlation coefficient (hv)".

P2, line 17: what did you mean with backscatter? Perhaps you meant back-scattered signal?

P2, line 17: I would delete "of hydrometeors" because for me it is redundant.

P2, lines 16-21: I suggest to say: "Dual-polarization radar provides characteristics of the precipitation such as precipitation type by means of . . . obtaining more informations about DSD (Cifelli et al., 2011), and reducing the impact of DSD variability on rainfall estimation. For these reasons rainfall estimates provided by polarimetric weather radar are better than that given by a single polarization weather radar.".

P2, line 25: I would remove "of the rain" because for me it is not necessary.

P2, line 27: please split the sentence into two sentences by replacing the word "and" by a mark.

P2, line 32: I would replace "with types of storms and stages of storm development"

with "with both types and stages of storm development".

P3, line 4: I would say "rainfall estimations by polarization radars are affected by errors due to different sources of uncertainties such as ... Reviews of the different sources of uncertainties are made by different Authors in the past, for example by Zawadzki (1984), Villarini and Krajewski (2010), Sebastianelli et al., 2013, and Spina et al., 2013, which I suggest the Authors to cite:

- Zawadzki, I.: Factors affecting the precision of radar measurements of rain, in: Proceeding of the 22d Conf. Radar Meteorology, Zurich, Switzerland, 10–13 September 1984, Amer. Meteor. Soc., 251–256, 1984.

- Villarini, G. and Krajewski, W. F.: Review of the different sources of uncertainty in single polarization radar-based estimates of rainfall, Surv. Geophys., 31, 107–129, 2010.

- Sebastianelli, S., Russo, F., Napolitano, F., & Baldini, L. (2013). On precipitation measurements collected by a weather radar and a rain gauge network. Natural Hazards and Earth System Science, 13(3), 605-623, doi:10.5194/nhess-13-605-2013.

- S. Spina, S. Sebastianelli, E. Ridolfi, F. Russo, L. Baldini, and L. Alfonso: Data selection to assess bias in rainfall radar estimates: An entropy-based method, AIP Conference Proceedings 1558, 1665 (2013); doi: 10.1063/1.4825849.

P3, line 5: please delete "These measurement errors affect rainfall estimation." Because for me it's redundant.

P3, line 6: I would replace "accurate measurement and calibration of ZH and ZDR" with ZH and ZDR" measurements.

P3, line 7: I would change "accommodation" in "assessment".

P3, line 9: I would replace "measured ZH and ZDR" in "ZH and ZDR measurements".

P3, lines 10-11: I suggest to use the wording "vertical profile of reflectivity" instead of

"measured ZH from the radar profiler (at vertical incidence)". See for example Sebastianelli et al., 2013.

P3, line 11: the disdrometer-inferred ZH.

P3, line 12: please change "by comparing reflectivity between radar and disdrometer" into "by comparison between radar and disdrometer reflectivity".

P3, line 19: the principle has been theorized by Gorgucci et al., 1992, please insert citation.

P3, lines 22-23: a mean axis ratio and a polarimetric rainfall algorithms.

P3, line 24: please remove "the" and change "after" in "hereafter".

P3, lines 25-26: please put into the brackets "and newly derived axis-ratio relation from 2DVD data".

P3, line 30: please remove "the" before "data".

P4, line 2: please replace "drawn" with "given".

P4, lines 6-9: please check the English grammar and rephrase. I would write "data were used to develop a mean raindrop axis ratio relation...", and "The disdrometer data used in this study were collected by a 2DVD from..."

P4, line 21: a frequency.

P4, line 23: I would write "Data are obtained by using six..."

P4, line 29: I would say "both for Zh and ZDR calibration".

P5, line 1: and for rainfall estimation.

P5, line 2: please replace "from" with "due to".

P5, line 7: please write "the 2DVD rainfall estimations" instead of "rainfall calculated from 2DVD data".

P5, lines 19-20: please rephrase. I suggest to say "In Fig. 2b and 2c we compare the axis ratio-diameter relation of Pruppacher and Beard (1970) with that found by the disdrometer before and after the correction, respectively.

P6, lines 12-13: the sentence is poorly written, please rephrase. I suggest to write "We analyzed the rainfall cases occurred during the period .... Fig. 3 shows six of these cases.

P6, line 18: As the 2DVD...

P6, line 25: the error is with respect to rain-gauge data? Please specify.

P7, line 1: you wrote "in suit". What did you mean? Perhaps you meant "on site"?

P7, lines 1-3: What's the criteria you used to distinguish between the different rainfall event types? Any references? In Sebastianelli et al., 2013 you can find a description of the rainfall events types from the radar point of view. Look also the works cited in that paper about that, in particular Zhang et al., 2008, and Zhang and Qi, 2010 which define events consisting of a stratiform part and a convective one (in relation to your table 2). So I suggest to add the following references:

- Sebastianelli, S., Russo, F., Napolitano, F., & Baldini, L. (2013). On precipitation measurements collected by a weather radar and a rain gauge network. Natural Hazards and Earth System Science, 13(3), 605-623, doi:10.5194/nhess-13-605-2013.

- Zhang, J. and Qi, Y.: A real-time algorithm for the correction of brightband effects in radar-derived QPE, J. Hydrometeorol., 11, 1157–1171, 2010.

- Zhang, J., Langston, C., and Howard, K.: Brightband identification based on vertical profiles of reflectivity from the WSR-88D, J. Atmos. Ocean. Tech., 25, 1859–1872, 2008.

P7, line 3: precipitation type.

P7, line 4: what did you mean with "difference rainfall"? Maybe you refer to the difference between disdrometer and rain gauge rainfall?

P7, line 7: I would change "larger" in "greater".

P7, line 17: you would write "reach about"?.

P7, line 18: established within.

P7, line 22: major and minor axis.

P8, line 1: 10.7 cm wavelength.

P8, line 2: we calculated.

P8, line 2: relations which.

P8, lines 15-16: you wrote "Polarimetric rainfall relations between R and dual-polarimetric parameters are derived when rain rate is greater than 0.1 mm hr-1.". Why? To avoid the smallest raindrop diameter and therefore the particles outliers? Please clarify.

P9, line 2: you wrote "The polarimetric radar contains systematic bias of the radar itself." It would be appropriate to introduce a bias definition. You can find a bias definition in Spina et al., 2013. In this work the calibration bias error is defined as a systematic error affecting radar estimates of rainfall independently from both the instant of measurement and the location of the sampling volume.

P9, line 3: I would suggest to replace "accomodation" with "assessment".

P9, line 4: I would suggest to write "the radar calibration" instead of "the calibration of the radar"; moreover, why the calibration was done only for light rainfall event? Maybe to avoid rain gauge error due to wind action which deflects the falling raindrops from the vertical, or it prevents the fall of the drops (updraft). So for the radar calibration you consider only stratiform cases? It is correct to say that?

P9, line 10: I would change "are point measurements" into "consist of point measurements".

P9, line 10: I would write "whereas radar data are measured in a sampling volume".

P9, line 18: I suggest the Authors to put into brackets "see Fig. 5".

P9, line 23: I suggest the Authors to write ".... Relation derived by measurements in the wind tunnel".

P9, lines 26-29: the sentence is too long and it is poorly written, please rephrase. I suggest to split it into two sentences. I suggest the Authors also to write "The Beard and Chuang (1987) polynomial relation (Eq. (9), black dashed line) is from 2.5 to 6.5 mm lower than the new mean axis-ratio relation values".

P10, lines 2-3: I would write "With the exception of this case, the new axis ratio was similar to Eq. (10) for diameters ranging from 3 to 5.5 mm".

P10, lines 5-6: please remove the sentence "This means that raindrops in South Korea are more oblate than the others"; it is my opinion that you can not assert this by comparing results of different models, which were obtained in different ways and with different data. To say something like that you should derive a mean axis-ratio relation with the same 2DVD in different part of the word.

P10, line 12: please change "from" in "by utilizing".

P10, line 13: I would write "and it was compared with R derived fron the 2DVD (Eq. (2))".

P10, line 14: the correlation coefficient. What correlation coefficient? Pearson correlation coefficient?

P10, line 15: I would write rain rate estimation as it is a disdrometer.

P10, line 19: observed rain rates.

P10, line 23: please check English grammar.
P10, line 24: I would change "when compared" in "in comparison".

P11, line 2: I would replace "from" with "measured by a".

P11, line 9: with the hourly rain rate.

P11, line 14: obtained by the radar.

P11, line 15: measured by the rain gauge.

P11, line 16: I would write "2DVD rainfall estimation".

P11, line 16: decide if you want to use showed or shown in the text.

P11, line 16: why good results. With respect to rain gauges? Please specify.

P11, line 17: I suggest the Authors to remove the sentence part "R(KDP, ZDR) > R(Zh, ZDR) > R(KDP) > R(Zh)", and to describe the concept just in words because in this way it seems that R(KDP, ZDR) is greater than rainfall estimated by each other formula. What did you mean is that the R(KDP, ZDR) algorithm give the most reliable rainfall estimation, it's right?

P11, lines 17-21: I would change "performed better" with "was more efficient"; it's not clear what did you mean when you write "on DSD statistics".

P12, line 2: I would replace "declined" with "worsen".

P12, line 4: for lower rain rates.

P12, lines 3-6: please delete "the radar rainfall estimations from" because it is redundant.

P12, lines 5-6: rainfall measured by gauges.

P12, line 8: I would write "the uncertainty in radar estimates due to the use of KDP reduces itself".

P12, line 18: ZH and ZDR biases were calculated separately for eight rainfall events.

So you calculated eight ZH biases and eight ZDR biases?

P12, line 22: the BSL ZDR measurements.

P12, lines 22-23: ZDR value is simulated by the 2DVD?

P12, line 25: please add the word "respectively" at the end of the sentence.

P12, line 27: please replace the word "comparing" with the word "comparison".

P12, line 30: I would write "... obtained before and after the bias correction, respectively, while ...".

P13, line 1: I would write "... obtained before and after the bias correction, respectively, while ...".

P13, lines 2-4: the sentence is not clear and poorly written. Please rephrase. Maybe you meant that before the bias correction the precipitation was 12.67 and after was 15.33? What's the corresponding value of the gauge?

P13, lines 4-5: The comparison with the rain gauge rainfall shows that after the bias correction the rainfall radar estimates were improved by about 13.71%, it's right?

P13, line 4: the rain gauge is the ground truth. You should say that also at the beginning of the paper (introduction and abstract).

P13, line 7: the sentence for me is poorly written, I would say "80.12 mm respectively for rain gauge, radar before bias correction, and radar after bias correction. The radar rainfall ...".

P13, lines 8-9: you calculated the bias for each event and then you performed a mean bias? It's right?

P13, line 9: MAE passes from.

P13, line 10: decreased from.
P13, line 10: please add "after the bias correction" at the end of the sentence.

P13, line 11: I would write "as well as MAE and RMSE values".

P13, line 12: I would say "both MAE and RMSE values decrease after correcting bias, and this means that rainfall estimation tended to improve after bias correction".

P13, lines 23-24: I would say "obtained through rainfall algorithms".

P13, line 24: I would say "was assessed by comparing 2DVD and BSL radar data with rain gauge measurements".

P13, line 25: I would change "was suited" into "is suitable".

P13, line 25: what's the meanings of "of the DSD statistics"? Maybe you intended "according to the DSD statistics"?

P13, line 26: please replace "had" with "have".

P13, line 29: I would change "was weak" into "is noisy". It's right?

P13, line 32: I would say "radar rainfall estimations was close to rain gauges measurements".

P14, line 1: I would say "different raindrops axis ratios". Did you mean the shape of the raindrops detected by the disdrometer?

P14, line 2: I would say ". . . relations, which were assessed to derive point . . .". P14, lines 3-4: I would say "obtained through rainfall algorithms".

P14, lines 4-5: the sentence is too long and it is not clear, please rephrase. I suggest the Authors to write: "Polarimetric algorithms will be developed to obtain areal rainfall estimation. A classification of rain rate based on them will be also performed in a future work".

P19, table 2: in the table caption please write "precipitation type" instead of "type of precipitation".

P19, table 2: what's represents PE[%]?

P22, table 5: I suggest the Authors to add two columns, the first one concerning the radar rainfall estimates, and the last one for rain gauge rainfall measurements.

P24, figure 2: you should clarify what represents the dotted line. Moreover you should specify that the red color correspond to the greater drop number density.

P26, figure 5: please add (b/a) at the y-label.

P31, figure 10: it's my opinion that it is better to replace the dotted lines with dashed lines.

---

## Author Comment (AC1) · 22 Mar 2016

**Response to the comments from the reviewer 1**

**Responses to the general comments:**

We would like to thank the reviewer for the constructive comments and corrections on the paper. The invaluable comments help improve our manuscript. Our responses are as follows.

In this document, for each comment (black font), we display an answer (blue font)

**Responses to the specific comments:**

*Q1) Page 3, ll.28-29: "Thereafter, improvement of quantitative rainfall estimation was investigated by applying derived calibration bias." This sentence is not clear. Please reformulate.*

A1) The sentence has been revised as follows:

*After*

"Improvement of quantitative rainfall estimation was investigated by applying calculated $Z_H$ and $Z_{DR}$ calibration bias of radar.

*Q2) Page 4, Section 2.1 (Disdrometer): It might be worth mentioning here that the 2DVD is considered one of the best and most reliable disdrometers on the market today.*

A2) I also think that 2DVD is the most reliable disdrometer compared to other disdrometers. However each instrument has different advantage and disadvantage. So, we did not include the above sentence.

*Q3) Page 5, Section 2.3 (Rain gauge): Please provide the brand/make and model number of the tipping bucket and specify if the data were quality controlled or not*

A3) The manuscript has been revised as follows:

"The rain gauge used in this study is a RG3-M tipping bucket rain gauge from Onset Computer Corporation. Maximum rainfall rate of rain gauge is 12.7 cm (5 in) per hour, and operating temperature range is from $0°$ to $50℃$. The bucket size of the rain gauge was 0.2 mm and time resolution was 0.5 s. The rain gauge is corrected to reduce instrumental uncertainty through field inter-comparison with reference gauge."

*Q4) Page 6, ll.10-11: Please explain why the rain rates from the 2DVD data are computed using the Brandes et al. (2002) velocity model while the older Atlas et al. (1973) velocity model is used to filter the 2DVD data.*

A4) Atlas et al. (1973) fall velocity relation is derived as an exponential formula, and Brandes et al. (2002) fall velocity relation is computed as a polynomial function. For this reason, Brandes et al. (2002) relation is widely used for calculation of rain rates from the 2DVD data.

A number of hydrometeor fall velocity outliers measured by the 2DVD. Some particles have velocities well beyond the terminal velocity ($\fallingdotseq$ 12 m/s) of large raindrops (Kruger and krajewski, 2002). So, we applied velocity-based filtering to reduce the effect of instrument errors. We use Atlas et al. (1973) fall velocity formula. This velocity relation has been widely used in many previous studies. In addition, the Atlas et al. (1973) velocity formula is used as a reference relation for comparison with measurement 2DVD data. Therefore, we use Atals et al. (1973) velocity model to filter the 2DVD data.

*Q5) Page 6, ll.24-25: "Therefore, the 2DVD data within 20%  error were used in this study." I'm not sure to fully understand what you mean by this. Are those 20% with respect to hourly accumulations or on an event basis?*

A5) The manuscript has been revised as follows:

*After*

"Therefore, the rainfall differences between 2DVD and rain gauge used in this study are limited to a maximum of 20% error, and the 2DVD data were excluded from the analysis when rainfall difference between 2DVD and rain gauge was exceeding 20%.

*Supplement*

According to previous studies, rainfall differences between disdrometer and rain gauge were mostly from 10% to 20%. Therefore, the rainfall differences between 2DVD and rain gauge used in this study are limited to a maximum of 20% based on these previous studies.

First, after finding all of the quality control of 2DVD and rain gauge data, we calculated rainfall differences (= percent error) for each rainfall events. Second, the 2DVD data were excluded from the analysis when rainfall difference between 2DVD and rain gauge was exceeding 20%. Each accumulated rainfall of rainfall events were used for calculation rainfall difference (= percent error). The difference rainfall of selected rainfall events are listed in Table 2.

*Q6) Page 7, ll.22-23: Here, it might be worth to say what you actually mean by "drop diameter" in this context. I assume you are referring to the diameter of a sphere with equal volume.*

A6) The manuscript has been revised as follows:

*Before*

"D is the raindrop diameter in mm"

*After*

"D is the equivalent volume diameter of the particle in mm. Hear, D is the diameter of a spherical drop of volume equal to the volume of the actual drop"

*Q7) Page 7, ll.27-28: Please provide at least one good reference for the T-matric method.*

A7) I have attached two references of the T-matric method.

Zhang, G., Vivekanandan, J., and Brandes, E.: A method for estimating rain rate and drop size distribution from polarimetric radar measurements, IEEE Trans. Geosci. Remote Sens., 39, 830-841, 2001.

Jung, Y., Zhangm G., and Xue, M.: Assimilation of simulated polarimetric radar data for a convective storm using ensemble kalman filter. Part I: Observation operators for reflectivity and polarimetric variable, Mon. Wea. Rev., 136, 2228-2245, 2008.

*Q8) Page 9, ll.1-2: "The polarimetric radar contains systematic bias of the radar itself." Not sure what you mean by this. Please reformulate.*

A8) The manuscript has been revised as follows:

*Before*

"The polarimetric radar contains systematic bias of the radar itself."

*After*

"The radar measurements are affected by various observational errors, such as ground echoes, beam broadening and abnormal propagation echoes, etc. In addition, calibration biases of radar $Z_H$ and $Z_{DR}$."

*Q9) Page 10, ll.5-6: "This means that raindrops in South Korea are more oblate than the others." This statement needs to be reformulated. There are many possible explanations for this and it would be premature to conclude that raindrops in South Korea are more oblate than in other places. The differences in axis-ratio might also be the result of instrumental effects, drop filtering and event selection. Please reformulate.*

A9) The manuscript has been revised as follows:

*After*

"These differences of raindrop shape can be caused by a variety of reasons, such as instrumental effects, fitting method, event selection, and different climatic regimes.

*Q10) Page 10, ll.20: "The correlation value of 0.10 mentioned in the text seems to be incorrect.*

A10) The manuscript has been revised as follows: 0.10 ⇒ 1.00

*Q11) Page 11, Eq.(12) and (13): There is no need to repeat the definition of the MAE and RMSE here.*

A11) First, we modified the Eq.(12) and Eq.(13) to Eq.(14) and Eq.(15).

R is the rain rate from observed one-minute 2DVD data in Eq (12) and (13) and, R is the averaged one-hour rain rate in Eq (14) and (15). So, we repeated the definition of MAE and RMSE to help readers understand better.

*Q12) Page 12, ll, 6-7: "In addition, the radar rainfall estimations from R(Kdp) and R(Kdp, Zdr) perform better than those of R(Zh, Zdr) for rain rates exceeding 5 mm/h". This is not obvious from the graph. Please provide hard evidence to back up this statement (e.g., in the form of an additional table or RMSE values for R>5mm/hr)*

A12) We tried to explain the phenomena $K_{DP}$ noise is reduced as rain rate increases (> 5 mm/hr), and combined polarimetric rainfall algorithm using $K_{DP}$ better than $R(Z_H)$ and $R(Z_H, Z_{DR})$ for estimated rainfall at higher rain rates (> 15 mm/hr). However, as your comment, this statement is misleading it was excluded from the paper. Also, the manuscript has been revised.

*Q13) Page 12, ll, 14-15: "Therefore, rainfall characteristics should be reflected in polarimetric rainfall relations." This is too vague. Please reformulate.*

A13) The manuscript has been revised as follows:

*After*

"Therefore, consideration of rainfall characteristics is necessary to improve the polarimetric rainfall algorithm."

*Q14) Page 20, Table 3: Please check if the low correlation values (0.10) are correct.*

A14) we change 0.10 to 1.00, thank you very much.

**Typos and English:**

We would like to express our sincere thanks to the reviewer for the positive encouragement to our work.

---

## Author Comment (AC2) · 22 Mar 2016

**Response to the comments from the reviewer 2**

**Responses to the general comments:**

We would like to thank the reviewer for the constructive comments and corrections on the paper. The invaluable comments help improve our manuscript. Our responses are as follows.

In this document, for each comment (black font), we display an answer (blue font)

**Responses to the specific remarks:**

*Q1) Page 6, ll.10: Why the Authors used the fall velocity-diameter relation of Brandes et al. (2002) to compute the rain rate and the one of Atlas et al. (1973) to filter out the outliers? Defend this choice or use the same relation.*

A1) Atlas et al. (1973) fall velocity relation is derived as an exponential formula, and Brandes et al. (2002) fall velocity relation is computed as a polynomial function. For this reason, Brandes et al. (2002) relation is widely used for calculation of rain rates from the 2DVD data.

A number of hydrometeor fall velocity outliers measured by the 2DVD. Some particles have velocities well beyond the terminal velocity ($\fallingdotseq$ 12 m/s) of large raindrops (Kruger and krajewski, 2002). So, we applied velocity-based filtering to reduce the effect of instrument errors. We use Atlas et al. (1973) fall velocity formula. This velocity relation has been widely used in many previous studies. In addition, the Atlas et al. (1973) velocity formula is used as a reference relation for comparison with measurement 2DVD data. Therefore, we use Atals et al. (1973) velocity model to filter the 2DVD data.

*Q2) Page 6, ll.12: How did the authors define an event? Please provide some information regarding the criterion used.*

A2) We have divided rainfall event into three rainfall types such as stratiform, convective and mixed rainfall events using the radar reflectivity and one-hour rain rate calculated by rain gauge and 2DVD. We were referring to Chang et al. (2009), and rainfall rate was modified. The criterion of the rainfall events are as follows:

|  | Stratiform | Convective | Mixed(Str+Con) |
|---|---|---|---|
| Reflectivity [dBZ] | $\leq 35$ | $> 35$ | - |
| Rainfall rate[mm/hr] | $\leq 5$ | $> 5$ | - |

Chang, W. Y., Wang, T. C., and Lin, P. L.: Characteristics of the Raindrop Size Distribution and Drop Shape Relation in Typhoon Systems in the Western Pacific from the 2D Video

Disdrometer and NCU C-Band polarimetric Radar, J. Atmos. Oceanic Technol., 26, 1973-1993, 2009.

Q3) Page 6, ll.26: How many events have been discarded by the quality control process?

A3) During the period from September 2011 to October 2012, total 38 rainfall events were measured by the 2DVD, five rainfall events were discarded by the quality control process. The excluded five rain events show that rainfall differences between 2DVD and rain gauge were mostly from 22% to 30%. A total of 33 rainfall events were analyzed and, the difference between 2DVD and rain gauge for the 33 rainfall events are listed in Table 2.

Q4) Page 7, ll.1-3: In the literature there are several different methods to distinguish between stratiform and convective rain. Can the Authors provide some information regarding the criterion used?

A4) As mentioned in A2), we have divided rainfall type into the stratiform and convective rain using the radar reflectivity and one-hour rain rate calculated by rain gauge and 2DVD. In addition, we distinguish precipitation type based on the dominant rainfall type over the observation time.

Q5) Page 7, ll.19: Can you provide some information regarding the choice of a third order polynomial? Did the Authors try also other relations (such as linear or a fourth order polynomial)? Can the Authors provide some information regarding the performance/goodness of the fitting (such as $R^2$)

A5) In order to produce the mean axis-ratio relation, a various fitting methods such as linear or polynomial (twice-, third-, fourth-order) fit were tried. As a result, the third-order polynomial relations were the most suitable for the observation data. For instance, as the raindrop size increased, the difference of raindrop size from the linear and twice-, fourth-order polynomial fit increased when compared with the 2DVD measurement data.

Q6) Page 8, ll.13: Can the Authors defend the choice of using Dmax=7 mm in the T-matrix simulation instead of using the maximum drop diameter measured by 2DVD in each DSD?

A6) The mean axis-ratio relation is necessary to calculate the complex scattering amplitudes of raindrops in the T-matrix simulation. This mean axis-ratio relation is based on 2DVD measurement data. Therefore, the effective diameter ($D_{max}$ = 7 mm) of mean axis ratio relation is used for simulation of T-matrix.

*Q7) Table 3: Please note that some correlation coefficients of Table 3 are very poor (0.01) please cheak this values.*

A7) we change 0.10 to 1.00.

*Q8) Page 10, ll.5-6: "This means that raindrops in South Korea are more oblate than the others." Please rewrite this sentence, it is "too strong". I think that there are not enough information to justify it.*

A8) The manuscript has been revised as follows:

*After*

"These differences of raindrop shape can be caused by a variety of reasons, such as instrumental effects, fitting method, event selection, and different climatic regimes.

*Q9) Page 10, ll.21: "Corr=0.01". Please check this value.*

A9) The manuscript has been revised as follows: 0.10 ⇒ 1.00

*Q10) Table 3: In terms of MAE, RMSE and corr., it seems that the use of different axis-ratio does not have a huge influence in particular the use of "new axis ratio" (that is optimized for the area) does not highly improve the three statistics. Can the Authors defend this issue?*

A10) Although, correlation coefficient according to the different axis ratio assumption are not significantly different, MAE and RMSE are showing differences in the polarimetric rainfall algorithms based on $Z_{DR}$ and $K_{DP}$ related to raindrop shape. These statistical differences can lead to a difference in coefficients of polarimetric rainfall relations. As your say, the use of new axis ratio does not highly improve the three statistics, however, the statistic of scatter showed the best result when using the $R(K_{DP}, Z_{DR})$, and the $R(K_{DP}, Z_{DR})$ based on new axis ratio showed the best performance.

*Q11) Page 11, ll.11: Why the statistical validation has been performed only on 18 events. How do you choses them?*

A11) We chose a continuous rainfall event for the continuity of measurement data. In other words, time of observation is short and unstable rainfall events were excluded from the analysis.

*Q12) Table 4: As for Table 3, also in this case the use of the "new axis ratio" does not produce a high improvement, in particular with respect to the relation of Pruppacher and Beard (1970). Please comment this issue.*

A12) First, axis ratio relation from Pruppacher and Beard (1970) used here is a linear relation, and new axis-ratio experimental fit is derived as third-order polynomial function. As you say, Table 4 show that, using the polarimetric parameter $K_{DP}$, the accuracy of the radar rainfall estimation is improved in rain estimation with the Pruppacher and Beard (1970) than that with the new axis ratio. According to Marzuki et al. (2013), "when inferring R from specific differential phase measured by dual-polarization radar, it is useful to have a linear equation between the mean axis ratio and drop diameter."

*Marzuki, M., Randeu, W. L., Kozu, T., Shimomai, T., Hashiguchi, H., and Schonhuber, M.: Raindrop axis ratios, fall velocities and size distribution over Sumatra from 2D-Video Disdrometer measurement, Atmos, Res., 119, 23-37, 2013.*

*Q13) Page 12, ll.1-2: the sentence is not clear to me. Which are the "DSD results"? How the Authors can say that, based on the results provided, "$K_{DP}$ is less sensitive to DSD variation and uncertainties in raindrop shape"? Please explain in more detail*

A13) The word "DSD results" means statistical results for 2DVD rainfall estimation, and we wanted to say that, the accuracy of the rainfall estimation improved when the $K_{DP}$ parameter was used for 2DVD rainfall estimation. However, as your comments, this is misleading, so sentence has been revised as follows:

*After*

"In the single rainfall relation with $Z_h$ (Figure 7a), an amount of scatter is present (MAE=0.95 mm h$^{-1}$ and RMSE=1.23 mm h$^{-1}$), and the scatter decrease (MAE=0.70 mm h$^{-1}$ and RMSE=0.92 mm h$^{-1}$) when the $K_{DP}$ parameter was used for 2DVD rainfall estimation (Figure 7b). These results are influenced by the variability of DSDs, and the effect of the DSD variability is declined in rainfall estimation with the R($K_{DP}$) or R($K_{DP}$, $Z_{DR}$) than that with the R($Z_h$)."

[Figure]

Figure 7. Scatter plot of one-hour rain rate from rain gauge (RG) and BSL S-Band radar (or 2DVD) based on Eq. (4) for 18 rainfall cases: The pluses represents one-hour gauge rain rate versus radar hourly rain rate from polarimetric rainfall algorithms, and squares indicate gauge and 2DVD rain rate by different polarimetric rainfall algorithms.

*Q14) Page 12, ll.7: I suggest to shows some plots regarding the performance of the algorithms for R > 5 mm/h*

A14) We tried to explain the phenomena $K_{DP}$ noise is reduced as rain rate increases (> 5 mm/hr), and combined polarimetric rainfall algorithm using $K_{DP}$ better than $R(Z_H)$ and $R(Z_H, Z_{DR})$ for estimated rainfall at higher rain rates (> 15 mm/hr). However, this sentence is misleading it was excluded from the paper. Also, the manuscript has been modified.

*Q15) Page 12, ll.9-10: This is known in the literature (such as Vulpiani et al. 2015 among others), please provide some reference.*

A15) I have attached one reference: Ryzhkov et al. (2005), Vulpiani et al. 2015

*Q16) Page 12, ll. 11-15: "The polarimetirc rainfall relations based on the new axis-ratio relation also were better than the others". Please explain how you can say that they are better. I think that this paragraph is about a crucial issue however it is not clear to me. I suggest to rewrite in order to explain the improvement of the new axis ratio in the rainfall retrieval.*

A16) The manuscript has been revised as follows:

When using the new mean axis-ratio relation, $R(K_{DP}, Z_{DR})$ is showing good results on 2DVD rainfall estimation and $R(Z_H, Z_{DR})$ is showing best performances on radar rainfall estimation. However, using the polarimetric parameter $K_{DP}$, the accuracy of the radar rainfall estimation is improved in rain estimation with the Pruppacher and Beard (1970) than that with the new axis ratio. According to Marzuki et al. (2013), when inferring rainfall from $K_{DP}$ measured by dual-polarization radar, it is useful to have a linear equation between the mean axis ratio and drop diameter. In addition, raindrop shapes are influenced by the temperature and pressure (Beard and Chuang 1987), and drop shape differences can be seen by the measurement errors, drop oscillation, dataset and fitting method (Thurai and Bringi 2005). Althought the difference in the value of the statistics seems small according to mean axis-ratio relations, it can lead to significant errors in the estimated DSD and rainfall rates (Bringi and Chandrasekar 2001). Therefore, consideration of rainfall characteristics is necessary to improve the polarimetric rainfall algorithm.

*Q17) Section 4.2.3: I think that the world "calibration" is misleading, the procedure explained in the paragraph is not really a calibration procedure, in my opinion it is more an adjustment of radar variables based on a ground truth (2DVD data). Pleas reformulate it. Moreover, the Authors did not compute the "calibration bias" for $K_{DP}$ why?*

A17) The reflectivity factor is affected by the absolute calibration error, and it require accurate knowledge of the radar constant. And differential reflectivity is independent of absolute radar calibration. These calibration errors can be calculated by various ways (Atlas 2002), it is commonly referred to as the "calibration of radar". In this study, we calculate daily $Z_h$ and $Z_{DR}$ calibration biases using the 2DVD data. This is adaptive calibration, in general the adaptive bias is more effective in terms of reduction of random error in rainfall estimation. In addition, the application of adaptive calibration biases is the most effective in reducing radar rainfall errors in particular for rainfall estimators with both $Z_H$ and $Z_{DR}$ (Kwon et al., 2015). $K_{DP}$ is independent of the absolute calibration error, attenuation because it is related to the phase shift of the electromagnetic wave. Therefore we did not compute the $K_{DP}$ calibration bias.

Atlas, D.: Radar calibration: some simple approaches, Bull. Amer. Meteor. Soc., 83, 1013-1316, 2002.

Kwon, S., Lee, G. W., and Kim, G.: Rainfall Estimation from an Operational S-Band Dual-Polarization Radar: Effect of Radar Calibration, J. Meteor. Soc. Japan., 93, 65-79,2015.

*Q18) Page 12, ll.18: Why do you use only 8 events for this analysis? Can you use all the 18 events selected for the validation? I noticed that in the 8 selected events there aren't convective events. Why?*

A18) To achieve accurate calibration bias of radar, the process of calibration bias should be performed in continuous and stable rainfall systems. Therefore, time of observation is short and unstable rainfall events were excluded.

*Q19) Page 12, ll.19-21: This is just a suggestion. Instead of doing two scatterplots for each event (namely Figure 8c and d for one event and Figure 9c and d for another one) the Authors can plot the results for all the events in two graphs (one for $Z_h$ and one for $Z_{DR}$)*

A19) I appreciate the suggestion. I'll take it into account.

**Editorial remarks:**

*Q1) Page 1, ll.16-17: "The shapes of raindrops have a direct impact on rainfall estimation". Although the sentence is correct in my opinion it is necessary to add that also the number, the dimension (diameter) and fall velocity of the raindrops play a central role.*

A1) The manuscript has been revised as follows:

*After*

"Polarimetric measurements are sensitive to size of raindrop, concentration, orientation and shape. Rainfall rates calculated from polarimetric radar are influenced by the shape of raindrop and canting. The shapes of raindrops play an important role in polarimetric rainfall algorithms based on differential reflectivity ($Z_{DR}$) and specific differential phase ($K_{DP}$). However, the characteristics of raindrop are different depending on precipitation type, storm stage of development, and regional and climatological conditions.

*Q2) Page 1, ll.20: I suggest to write "raindrop size distribution". Instead of "raindrop shapes"*

A2) We change "raindrop size distribution" to raindrop shapes

*Q3) Page 2, ll.2: what does it means "DSD statistics"? There worlds are used several times in the manuscript, but it is not clear the exact meaning. Please clarify it.*

A3) "DSD statistics" means the statistical results of the precipitation derived from the 2DVD data.

*Q4) Page 2, ll.24-25: "This is because the shape of raindrops is one of the most sensitive parameters for representing the DSD properties of the rain". Please consider to reformulate this sentence or to put one or more reference. There are other important parameters (e.g. diameter, number of drops etc) that should be mentioned.*

A4) The manuscript has been revised as follows:

*After*

"Polarimetric radar measurements are sensitive to the DSD properties such as diameter, concentration, orientation, and shape. Rainfall rates derived from polarimetric radar measurements are affected by the mean shape of raindrops and canting (Brandes et al., 2002).

*Q5) Page 3, ll.7-10: Please check this sentence, single polarimetric radar does not provide $Z_{dr}$*

A5) we change "single polarimetric radar" to "polarization radar"

*Q6) Page 3, ll.24: I think that "four" should be substitute to "three"*

A6) we change "four" to "three"

*Q7) Page 6, ll.3-4: This sentence should be moved to page 5 line 25 after "...Atlas et al. 1973)". Furthermore it is not clear the meaning of "the normal distribution". Please reformulate it.*

A7) As your comments, "The value outside…" sentence was moved to page 5 line 25, and "the normal distribution….." sentence has been revised as follows:

*Before*

"Some of the outliers of fall velocity and oblateness distribution were beyond the normal distribution"

*After*

"Some particles have fall velocities beyond the terminal velocity of large raindrops.

*Q8) Page 7, ll.13-14: similar to my previous comments, I think that the sentence "this relation is one of the most sensitive parameters for representing the rainfall properties" is "too strong". Can you provide some references or reformulate it?*

A8) The manuscript has been revised as follows:

*After*

"The mean raindrop shape is related to the precipitation type, regional and climatological conditions, and it affects rainfall rates derived from polarimetric radar measurements.

*Q9) Table 5: check the format of the date.*

A9) We have modified the Table 5

**References:**

Vulpiani, G., L. Baldini, and N. Roberto. "Characterization of Mediterranean hail-bearing storms using an operational polarimetric X-band radar." Atmospheric Measurement Techniques 8.11(2015): 4681-4698

---

## Author Comment (AC3) · 22 Mar 2016

**Response to the comments from the reviewer 3**

**Responses to the general comments:**

We would like to thank the reviewer for the constructive comments and corrections, especially about the description of the polarimetric rainfall algorithms. The invaluable comments help improve our manuscript. Our responses are as follows.

In this document, for each comment (black font), we display an answer (blue font)

**Responses to the specific comments:**

*Q1) Page 1, ll.19-21:It is not clear the link of the sentence starting with "therefore" with the previous one*

A1) As your comments, the manuscript has been revised as follows:

*After*

"Polarimetric measurements are sensitive to size of raindrop, concentration, orientation and shape. Rainfall rates calculated from polarimetric radar are influenced by the shape of raindrop and canting. The shapes of raindrops play an important role in polarimetric rainfall algorithms based on differential reflectivity ($Z_{DR}$) and specific differential phase ($K_{DP}$). However, the characteristics of raindrop are different depending on precipitation type, storm stage of development, and regional and climatological conditions.

*Q2) Page 2, ll.16-18: Potential and actual advantages of polarimetry applied to radar systems have been analyzed in the literature. However, this sentence is not very precise.*

A2) As your comments, we added the information of dual-polarization radar measurement, and we have modified the sentence as follows:

*Added the sentence*

"...because, more information about raindrop size distribution (DSD) is available, and dual-polarization radar can distinguish precipitation type."

*Modified the sentence*

*Before*

"Dual-polarization radar provides characteristics of the precipitation by backscatter and differential propagation phase of hydrometeors, and therefore can obtain more information

about DSD"

*After*

"Dual-polarization radar provides characteristics of the precipitation by backscatter and differential propagation phase of hydrometeors and therefore can reveal uncertainty of rainfall estimation resulting from DSD variability"

*Q3) Page 2, ll.20-21: Dual polarization rainfall algorithms used $K_{DP}$ or combinations of Zh, Zdr, and $K_{DP}$.*

A3) The manuscript has been revised as follows:

*After*

"Therefore, dual-polarization rainfall algorithms used $K_{DP}$ or combinations of $Z_H$, $Z_{DR}$, and $K_{DP}$ are better than using reflectivity factor only."

*Q4) Page 2, ll.24-25: This is wrong: maybe "allowing correct interpretation of polarimetric measurements in rain" or "an important feature of rain microphysics".*

A4) The sentence on ll.24-25 ("This is because…of the rain") is misleading, so it was excluded from the paper. Instead, we added the following sentence

*Added sentence*

"Polarimetric radar measurements are sensitive to the DSD properties such as diameter, concentration, orientation, and shape. Rainfall rates derived from polarimetric radar measurements are affected by the mean shape of raindrops and canting (Brandes et al., 2002).

*Q5) Page 2, ll.26-28: Authors should cite also more recent work on the drop-shape topic.*

A5) As your comments, we added the recent reference (Brandes et al., 2002; Thural and Bringi, 2005; Marzuki et al., 2013)

*Q6) Page 3, ll.1-3: "Thus,.."Again, I do not see how this conclusion follows from the previous sentence.*

A6) As your comments, we have modified the manuscript.

*After*

"However, they are not frequently studied in Korea. Therefore, the shape of raindrop and polarimetric rainfall algorithm reflecting rainfall characteristics of the Korean peninsula studies are necessary to improve rainfall estimation."

*Q7) Page 3, ll.14-17: Cited studies by Goddard et al. aimed at demonstrating the inadequacy of the Pruppacher and Beard shape-size relation.*

A7) Goddard et al. (1982) suggest that the raindrop axis ratio relation from Pruppacher and Beard (1973) needs careful consideration. But, the overall agreement between the radar and disdrometer measurements was generally good.

*Q8) Page 3, ll.19: Formulation of self-consistency by Gorgucci et al. (1992) was based on Zh, $Z_{DR}$ and $K_{DP}$ and not on Zh-$K_{DP}$.*

A8) The calibration bias of radar reflectivity is calculated from the comparison of measured $\Phi_{DP\_obs}$ with calculated $\Phi_{DP\_cal}$ derived from $Z_h$ using the $Z_h$-$K_{DP}$ self-consistency relationship as following procedure (Lee and Zawadzki 2006).

1. Select the rain region to avoid the ground echoes and bright band contamination.

2. Calculate the $K_{DP}$ at the rain region from the observed $Z_h$ using the $Z_h$-$K_{DP}$ relationship

3. Calculate $\Phi_{DP\_cal}$ by integrating the calculated $K_{DP}$ along the ray.

4. Find $\Phi_{DP\_obs}$ measured in the same ray.

5. Calculate the calibration bias by comparing $\Phi_{DP\_cal}$ and $\Phi_{DP\_obs}$.

Lee, G., and Zawadzki, I.: Radar calibration by gage, disdrometer, and polarimetry: Theoretical limit caused by the variability of drop size distribution and application to fast scanning operational radar data, J. Hydrol., 328, 83-97, 2006.

*Q9) Page 3, ll.21: A recent paper by Chandrasekar et al. (2015) is more appropriate and update than Atlas et al. (2002).*

A9) I appreciate your advice, after examining Chandrasekar et al. (2015), I'll consider your suggestion.

*Q10) Page 3, ll.24: Replace "four" with "three"*

A10) we change "four" to "three"

*Q11) Page 4, ll.20: Please add the height ASL of the radar*

A11) Sea level of antenna of BSL radar is 1,085 m, we added the ASL of radar.

*Q12) Page 5, ll.1-2: A 0-deg elevation allows small distances between measurements aloft and ground measurements. For most installations measurements collected at such small elevation angles are prone to effects of nearby obstacles. Please demonstrate that for the Bislan radar, this elevation does not implies beam blocking/ground clutter effects or that they are negligible.*

A12) 2DVD data are ground measurements and radar data are volume measurements. To compare polarimetric radar parameters, it is necessary to minimize the influence of height difference of 2DVD and radar, and effect by ground. If using high elevation, we can avoid effects from beam blocking and ground echoes on the measurements, however, this elevation is a very great difference in measurement height. Figure 1 show beam path of the BSL radar and 2DVD location. The 2DVD is located about 22.3 km (17°) away from the BSL radar. The 0.0° PPI radar data can avoid effect from beam blocking and ground echoes. Thus, the 0.0° PPI radar data were used.

[Figure]

Figure 1. Beam path and terrain map in 17° azimuth angle of the BSL radar.

*Q13) Page 5, ll.2-5: Please explain how averaging of PHIdp is obtained.*

A13) We were wrong description, so the manuscript has been revised as follows. $\Phi_{DP}$ is measured by radar, and $K_{DP}$ was calculated from the filtered $\Phi_{DP}$ ($\Phi_{DP}$ unfolding and FIR (Finite Impulse Response) filter were applied to the $\Phi_{DP}$ measurement data)

*Before*

"The $Z_H$, $Z_{DR}$, $\Phi_{DP}$, and $\rho_{hv}$ radar parameters were averaged …"

*After*

"The $Z_H$ and $Z_{DR}$ radar parameters were averaged …"

*Q14) Page 5, ll.21: What does it mean "beyond the normal distribution"?*

A14) "beyond the normal distribution….." sentence has been revised as follows:

*Before*

"Some of the outliers of fall velocity and oblateness distribution were beyond the normal distribution"

*After*

"Some particles have fall velocities beyond the terminal velocity of large raindrops.

*Q15) Page 6, ll.10-11: Why do Authors use the velocity-size relation by Brandes et al. (2002) and Atlas et al. (1973) to filter 2DVD measurements? Note that, starting from 2DVD counts, such relation is not necessary for computing R.*

A15) Atlas et al. (1973) fall velocity relation is derived as an exponential formula, and Brandes et al. (2002) fall velocity relation is computed as a polynomial function. For this reason, Brandes et al. (2002) relation is widely used for calculation of rain rates from the 2DVD data.

A number of hydrometeor fall velocity outliers measured by the 2DVD. Some particles have velocities well beyond the terminal velocity ($\fallingdotseq$ 12 m/s) of large raindrops (Kruger and krajewski, 2002). Therefore, we applied velocity-based filtering to reduce the effect of instrument errors. We use Atlas et al. (1973) fall velocity formula. This velocity relation has been widely used in many previous studies. In addition, the Atlas et al. (1973) velocity formula is used as a reference relation for comparison with measurement 2DVD data. Therefore, we use Atals et al. (1973) velocity model to filter the 2DVD data.

*Q16) Page 7, Section "Raindrop axis ratio": There are a number of questions about the fitting (4). First, what is the accuracy of the fitting. Second, is this fitting more appropriate for certain events (i.e. is there an event-by-event variation?).*

A16) · *First, what is the accuracy of the fitting* ⇒ We derived new raindrop axis ratio relation using 2DVD measurement. For reliability of the 2DVD data, we apply to the velocity-based filter for remove drop outliers (Thurai and Bringi, 2005), and compared the rain rate calculated from the 2DVD data to collocated rain gauges measurement. In addition, the oblateness data corresponding to raindrop diameters smaller than 0.5 mm were removed when we derived the new axis-ratio relation. Also, although the measured maximum diameter from the 2DVD could reach about 8.0 mm, the fitting was established to within 7 mm in order to obtain accurate information from the appropriate data.

In order to produce the mean axis-ratio relation, a various fitting methods such as linear and polynomial (twice-, third-, fourth-order) fit were tried. As a result, the third-order polynomial relation was the most suitable for the observation data. For instance, as the raindrop size increased, the difference of raindrop size from the linear and twice-, fourth-order polynomial fit increased when compared with the 2DVD measurement data.

· *Second, is this fitting more appropriate for certain events* ⇒ The mean raindrop axis-ratio relation is based on measurement data collected by 2DVD, a total of 33 rainfall events were used for deriving the empirical relation. The dataset consisted of 15 stratiform rainfall events, 12 convective rainfall events, and 6 mixed (str/con) rainfall events with 17,618 min DSD samples. The majority of rainfall events have 0-5 mm raindrop diameter. Comparison results according to rainfall type, the derived relation was appropriate in all rainfall types. There are no significant differences between the rainfall types.

However, Marzuki et al. (2013) derive axis-ratio relation for stratiform, mixed stratiform/convective and shallow convective. According to paper, axis ratio of deep convective is slightly larger than for other rain types.

*Marzuki, M., Randeu, W. L., Kozu, T., Shimomai, T., Hashiguchi, H., and Schonhuber, M.: Raindrop axis ratios, fall velocities and size distribution over Sumatra from 2D-Video Disdrometer measurement, Atmos, Res., 119, 23-37, 2013.*

*Q17) Page 7, Section "Disdrometer-rainfall algorithms"*: I think it is more appropriate to $Z_h$, $Z_{DR}$, $K_{DP}$ "variables" or "measurements" instead of "parameters"

A17) As your comment, we change "parameters" to "variables"

*Q18) Page 8, ll.12: Likely values of mean and standard deviation are switched.*

A18) I'd appreciate your input on this. The manuscript has been revised as follow:

*After*

"The terms $\overline{\emptyset}$ and σ are assumed to be 0° and 7°, respectively.

*Q19) Page 9, Section 3.4: What is the point of using light rain? $Z_{DR}$ near zero?. Please explain. What is the accuracy expected with this calibration?*

A19) We chose a continuous rainfall event for the continuity of measurement data, and we also use stable light rainfall event in order to avoid the impact of the unstable rain (e.g. convective, short observation of time of storms, beginning or an end of the storms).

The BSL radar measured $Z_H$ and $Z_{DR}$ can be compared with theoretical $Z_H$-$Z_{DR}$ relation (derived by disdormeter) to verify the measurement data. Comparison result of $Z_{DR}$ using the theoretical $Z_H$-$Z_{DR}$ relation, the measured $Z_H$ and $Z_{DR}$ showed a substantially lower than theoretical $Z_H$ and $Z_{DR}$. Thus, the determination of the calibration bias of $Z_H$ and $Z_{DR}$ is essential to improve the accuracy of radar rainfall estimation.

*Q20) Page 10: "Variability of DSD in rainfall estimation" I would like to see also some relative performance factors, such as the ratio of RMSE and average value of R.*

A20) To investigate the variability of DSD in rainfall estimation, R derived from observed DSDs of 17,618 min are compared with $R_e$ estimated from combinations of polarimetric measurement (Figure 6). All derived relationships and statistics are shown in Table 3. The mean absolute error (MAE), the root-mean-square error (RMSE), and correlation coefficient (Corr.) are used for evaluating the DSD variability. And the 17,618 min DSD data used here is 2DVD measurement data during 2011 to 2012.

*Q21) Page 10, ll.21: Corr=0.1??*

A21) The manuscript has been revised as follows: 0.10 ⇒ 1.00

*Q22) Page 11, ll.3:"A summary of…" The beaviour of the different algorithms with intrinsic dual-pol measurements is what is expected (e.g. Bringi and Chandrasekar, 2001). What is strange is that only $R(K_{DP}, Z_{DR})$ takes advantage from the new shape-size relation.*

A22) ·First ⇒ As your comments, we added the following sentence and reference. "The

reflectivity factor is affected by the absolute calibration error, and it require accurate knowledge of the radar constant. The differential reflectivity is independent of absolute radar calibration. Therefore it can be measured without begin affected by absolute calibration errors. However, $Z_{DR}$-based algorithm needs to be used in conjunction with $Z_h$ or $K_{DP}$, because $Z_{DR}$ is a relative power measurement. Unlike $Z_h$ and $Z_{DR}$, $K_{DP}$ is independent of the absolute calibration error, attenuation because it is related to the phase shift of the electromagnetic wave. However, $K_{DP}$ is relatively noisy in light rain (low rain rate). Thus, the pros and cons of each polarimetric variable translate into the error of rainfall algorithms (Bringi and Chandrasekar, 2001)."

·Second $\Rightarrow$ $R(K_{DP}, Z_{DR})$ is showing good results on 2DVD rainfall estimation when using new mean axis-ratio relation. These results are influenced by the variability of DSDs, and the effect of the DSD variability is declined in rainfall estimation with the $R(K_{DP})$ or $R(K_{DP}, Z_{DR})$ than that with the $R(Z_h)$. But, the accuracy of the rainfall estimation declined when the $K_{DP}$ parameter was used for radar rainfall estimation. Whereas $R(Z_H, Z_{DR})$ is showing best performances on radar rainfall estimation. This was because $K_{DP}$ measured by radar is noisy in low rain rate.

Using the polarimetric parameter $K_{DP}$, the accuracy of the radar rainfall estimation is improved in rain estimation with the Pruppacher and Beard (1970) than that with the new axis ratio. According to Marzuki et al. (2013), when inferring rainfall from $K_{DP}$ measured by dual-polarization radar, it is useful to have a linear equation between the mean axis ratio and drop diameter. In addition, raindrop shapes are influenced by the temperature and pressure (Beard and Chuang 1987), and drop shape differences can be seen by the measurement errors, drop oscillation, dataset and fitting method (Thurai and Bringi 2005).

*Q23) Page 11, ll.21-22: now, it is $R(Z_H, Z_{DR})$ the best algorithm and is the only one that take advantage from new shape-size relation. This is also not surprisingly. A simple exercise consisting in adding a properly modeled error to intrinsic measurements would reveal how algorithms are sensitive to random measurement fluctuation and/or calibration biases (see again Bringi and Chandrasekar, 2001). The bad performance of $R(K_{DP}, Z_{DR})$ can be ascribed to the an unappropriate $K_{DP}$ estimation (see the increase in the error of radar $R(K_{DP})$. From figure 7 compared with figure 6, I would expect worst MAE and RMSE values than those in Table 3. Finally, it is not clear to me whether Zh and/or $Z_{DR}$ bias correction was applied or not here.*

A23) The invaluable comments help improve our manuscript. However, the purpose of this study was to examine the performance of polarimetric rainfall algorithm according to raindrop shapes. So, various errors such as radar measurement errors and error due to the parametric form (R) were equally applied to the algorithms. I appreciate the suggestion and we'll consider it in future research. And Table 4 is the uncorrected result, and the Table 5 shows the results of applying the daily $Z_H$ and $Z_{DR}$ biases.

*Q24) Page 12, Section: "Correction of calibration bias": What is the accuracy of this calibration? Can the event-to event variability of the bias be related to variation of radar performance? Figure 9 shows clearly that the estimation of Zdr bias is extremely poor.*

A24) Adaptive calibration is daily $Z_h$ and $Z_{DR}$ calibration biases, and the verification of rainfall estimation is performed by applying adaptive calibration biases that vary each rain event. In general the adaptive bias is more effective than the averaged bias in terms of reduction of random error in rainfall estimation. In addition, the application of adaptive calibration biases is the most effective in reducing radar rainfall errors in particular for rainfall estimators with both $Z_H$ and $Z_{DR}$ (Kwon et al., 2015).

The rainfall event on 23 August 2012 (Figure 9) is mixed rainfall event, and $Z_{DR}$ biases in weak reflectivity were lower than biases in strong reflectivity. Therefore, low bias seems to be derived because daily $Z_{DR}$ bias is calculated by average during the observation time.

**References**

E. Gorgucci, G, Scarchilli, and V. Chandrasekar, Calibration of radar using polarimetric techniques, IEEE Trans. Geosci. Remote Sens., vol. 30, no. 5, pp. 853-858, Sep. 1992

Chandrasekar V., Baldini, L., Bharadwaj N., Smith, P. L., Calibration Procedures for Global Precipitation Measurement Ground-Validation Radars, TheRadio Science Bulletin No 355 (December 2015), pp. 45-73

[http://www.ursi.org/files/RSBissues/RSB_355_2015_12Corrected.pdf]

V.N.Bringi and V. Chandrasekar, Polarimetric Doppler Weather Radar: Principles and Applications. Cambridge, U.K.:Cambridge Univ. Press, 2001, p. 648

---

## Author Comment (AC4) · 22 Mar 2016

**Response to the comments from the reviewer 4**

**Responses to the general comments:**

We would like to thank the referees for their helpful comments, especially about the description of the motivation and raindrop shapes. These comments will help to improve considerably the quality of the paper. Our responses are as follows.

In this document, for each comment (black font), we display an answer (blue font)

**Responses to the specific comments:**

*Q1) Page 2, ll.29: I don't agree that "[t]he raindrop shape is defined by the shape-size relationship of raindrop." The shape is rather defined by interactions between the drop and the atmosphere (and other drops). However, the average shape of a raindrop can be inferred from its size.*

A1) The sentence modified as suggested in your comment:

*After*

"The average shape of a raindrop can be inferred by the shape-size relationship of a raindrop."

*Q2) Section 2.1: Additional information about the siting of the 2DVD should be given here: things like closeness to buildings could be significant.*

A2) The sentence modified as suggested in your comment:

*After*

"The disdrometer data used in this study were measured using a 2DVD, and the data were collected from September 2011 to October 2012 deployed in the campus of Kyungpook National University (KNU), Daegu, Korea (35.9°N, 128.5°E). The 2DVD used in this study is compact 2DVD version, which installs in observation field from the buildings."

*Q3) Page 5, ll.1: You say that you used the 0.0° elevation "to avoid effects from beam blocking and ground echoes". Surely that would be further reduced by using the 1.6° elevation data? you should say why you opted for 0.0° rather than 1.6° (to have measurements as close as possible in space to the 2DVD?).*

A3) 2DVD data are ground measurements and radar data are volume measurements. To compare polarimetric radar parameters, it is necessary to minimize the influence of height difference of 2DVD and radar, and effect by ground. If using 1.6° elevation, we can avoid effects from beam blocking and ground echoes on the measurements, however, this elevation is a very great difference in measurement height. Figure 1 show beam path of the BSL radar and 2DVD location. The 2DVD is located about 22.3 km (17°) away from the BSL radar. The 0.0° PPI radar data can avoid effect from beam blocking and ground echoes. Thus, the 0.0° PPI radar data were used.

[Figure]

Figure 1. Beam path and terrain map in 17° azimuth angle of the BSL radar.

*Q4) Figure 3: You seem to be showing some of the best available cases. Better might be to show a representative sample, including some of the best and some of the worst.*

A4) During the period from September 2011 to October 2012, total 38 rainfall events were measured by the 2DVD, five rainfall events were discarded by the quality control process. The excluded five rain events show that rainfall differences between 2DVD and rain gauge were mostly from 22% to 30%. In this study, we did not include five rainfall events. The difference between 2DVD and rain gauge for the 33 rainfall events are listed in Table 2.

*Q5) Section 3.2: You should give more information on the dataset you were working with here. What size of diameter bins were you using, what was the minimum number of drops in a bin included in the final fitting, what were the axis ratio standard deviations, etc.?*

A5) The Section 3.2 modified as suggested in your comment.

*After*

"Size of diameter bin is 0.2 mm, and the oblateness data corresponding to raindrop diameter smaller than 0.5 mm were removed when we derived the mean axis-ratio relation, because oblateness is relatively noisy in small raindrop. In addition, although the measured maximum diameter from the 2DVD could reach about 8.0 mm, the fitting was established to within 7 mm in order to obtain accurate information from the appropriate data.

In order to produce the mean axis-ratio relation, a various fitting methods such as linear and polynomial (twice-, third-, fourth-order) fit were tried. As a result, the third-order polynomial relation was the most suitable for the observation data."

*Q6) Page 8, ll.12: it may just be typo, but I think you should be using a mean canting angle of 0° and a standard deviation of 7°, not the other way round.*

A6) I'd appreciate your input on this. The manuscript has been revised as follow:

*After*

"The terms $\overline{\emptyset}$ and $\sigma$ are assumed to be 0° and 7°, respectively.

*Q7) Section 3.4: it is not quite clear here whether you are applying these calibration corrections only in table 5, or also in table 4. If in table 4, then assuming the new axis ratio calibration values in all cases would perhaps unfairly disadvantage the other axis ratios considered. Additionally, you should explain why you consider only light rain events for the calibration.*

A7) The calibration bias of $Z_H$ and $Z_{DR}$ were applied in Table 5, we did not apply calibration bias in Table 4. Therefore, accuracy of rainfall estimation is evaluated as fair. In addition, to achieve accurate calibration bias of radar, the process of calibration bias should be performed in continuous and stable rainfall systems. Because in order to avoid the impact of the unstable rain (e.g. convective, short observation of time of storms, beginning or an end of the storms). Therefore, time of observation is short and unstable rainfall events were excluded.

*Q8) Section 4.2.1: I think that here Re is begin determined by applying the various algorithms to radar polarimetric variables derived from the 2DVD data. is that correct? Either way, you should make this clearer.*

A8) Yes, $R_e$ is rain rate from various polarimetric rainfall algorithms. $R_e$ is then obtained from the 2DVD data. The sentence modified as suggested in your comment.

*Q9) Table 3: For the $K_{DP}$, $Z_{DR}$ algorithms, I suspect the 0.1 correlation values are wrong (compare with figure 6)*

A9) The manuscript has been revised as follows: 0.10 ⇒ 1.00

*Q10) Section 4.2.2: Why do you carry out this validation only on 18 of the rain events and not all 33?*

A10) We chose a continuous rainfall event for the continuity of measurement data. In other words, time of observation is short and unstable rainfall events were excluded from the analysis.

**Responses to the technical corrections:**

*Q1) Page 1, ll 26: I think you mean different not differences here.*

*Q2) Page 6, ll 10: I think you mean that the drop fall velocity formula used was that derived by Brandes et al. (2002)*

*Q3) Page 7, ll 1: in situ rather than "in suit"*

A) We would like to express our sincere thanks to the reviewer for the positive encouragement to our work.

**References:**

Thurai, M. et al.: "Drop Shapes, Model comparisons, and calculations of polarimetric radar parameters in rain", J. Atmos. Oceanic Technol. 24, 1019-1032, June 2007

Gorgucci, E., Chandrasekar, V. and Baldini, L.: "Can a unique model describe the rain-drop shape-size relation" A clue from polarimetric radar measurements?", J. Atmos and oceanic Technol. 26, 1829-1842, September 2009

Marzuki, Randeu, W.L., Kozu, T., Shimonai, T., Hashiguchi, H. and Schonhuber, M.:"Raindrop axis ratios, fall velocities and size distribution over Sumatra from 2D-Video Disdrometer measurement", Atmos. Res, 119, 23-37, January 2013

---

## Author Comment (AC5) · 24 Mar 2016

**Response to the comments from the reviewer 5**

**Responses to the general comments:**

We would like to thank the reviewer for the constructive comments and corrections, especially about the English language. The invaluable comments help improve our paper. Our responses are as follows.

In this document, for each comment (black font), we display an answer (blue font)

**Responses to the specific comments and Technical corrections:**

*Q1) Page 1, title: I would suggest "Dual-polarization radar rainfall estimation in Korea according to raindrop shapes obtained by a 2D Video Disdrometer".*

A1) I'll take it into account

*Q2) Page 1, ll.23: What did you mean with "a new axis ratio of raindrop relations"? it's a relation between the raindrop axis ratio and the raindrop diameter? Please specify. It's my opinion that you should better introduce this relation both in the abstract and in the introduction.*

A2) The average shape of a raindrop can be inferred by the shape-size relationship of a raindrop. The equilibrium raindrop shapes are defined by the relation between the raindrop axis ratio versus equivolume diameter D [mm]. As your comments, we added the information of axis ratio relation.

*Q3) Page 1, ll.24-25: If you say: "polarimetric rainfall algorithms were derived using different axis ratio relations", it seems that the polarimetric algorithm changes depending on the axis ratio relation you consider. But if we observe table 3 you applied each polarimetric rainfall relation with the same 4 axis ratio relations. Instead, the polarimetric parameter depend on the shape of raindrops you assumed, as it's better specify at page 7 lines 28-29. So you should rephrase.*

A3) Each theoretical polarimetric variables were simulated according to different mean axis-ratio relations (the shape of the raindrops), from them, polarimetric rainfall algorithms were derived. As your comments, manuscript has been revised.

*Q4) Page 1, ll.25-26: What did you mean saying "radar-point one hour rain rate"? A radar estimates rain in a pixel not in a point. A rain gauge measures rain in a point. Could you specify if the pixel in question includes within it the rain-gauge location? Furthermore with "one-hour rain rate" did you mean the average rain rate in a hour? It would be better if you put the unit. I would suggest to write "the radar-pixel hourly mean rain rate".*

A4) In this study, one-hour rain totals that were obtained from the radars and gauges were compared. We examined "point" estimates of the one-hour rain accumulation. By point estimate we mean an hourly total averaged over a small (1km x 2°) area centered on a rain gauge.

*Q5) Page 1, ll24-27: you wrote: "Second, polarimetric rainfall algorithms were derived using different axis ratio relations, and estimated radar-point one hour rain rate for the differences in polarimetric rainfall algorithms were compared with the hourly rain rate measured by gauge." The sentence is too long, please split it. Moreover, I suggest writing: "the estimated radar-pixel hourly mean rain rate obtained from the different polarimetric rainfall algorithms" instead of "estimated radar-point one hour rain rate for the differences in polarimetric rainfall algorithms"*

A5) As your comments, we split the sentence, and the manuscript has been revised as follow:

*After*

"Second, polarimetric rainfall algorithms are derived using different axis ratio relations. The estimated radar-point hourly mean rain rate obtained from the different polarimetric rainfall algorithms are compared with the hourly rain rate measured by a rain gauge."

*Q6) Page 1, ll.27-28: I would change "in relation to calibration bias of reflectivity and differential reflectivity" into "to calibrate reflectivity and differential reflectivity biases".*

A6) I'd appreciate your input on this. The manuscript has been revised

*Q7) Page 1, ll.28-29: the sentence is poorly written, I suggest to write: "For D <1.5 mm and for D >5.5 mm the shape of raindrops obtained by the new axis-ratio relation developer by the 2DVD is more oblate than the shapes obtained by the existing relations."*

A7) As your comments, the sentence has been revised

*Q8). Page 2, ll.6-9: the sentence is poorly written, I would write "Zh and ZDR biases were calculated by comparing Zh and ZDR radar measurements with the same parameters simulated by the 2DVD. In order to produce more accurate rainfall estimation." (it is redundant to say that a bias is used to correct a bias). In addition it's my opinion that you should better specify what is the aim of your work. If I understood, you developed an new axis-ratio relation with the 2DVD. You developed also the polarimetric rainfall algorithms by using the same tool? Then, you applied polarimetric rainfall algorithms to calculate the rain by utilizing as inputs the polarimetric parameters, which are both measured by the radar and simulated by the disdrometer, according to 4 axis-ratio relations. You corrected also Zh and ZDR biases by comparing radar measurements of polarimetric parameter with the disdrometer simulations of the same parameters. Finally you validated the rainfall estimations obtained by both the radar and the disdrometer by a comparison with the rain gauges measurements considered as the ground truth. In many past works the rain gauge observations have been considered as the ground truth, for example in both Lombardo et al. (2006) and Sebastianelli et al. (2013) which I suggest the authors to cite:*

*- Lombardo, F., Napolitano, F., Russo, F., Scialanga, G., Baldini, L., and Gorgucci, E.: rainfall estimation and ground clutter rejection with dual polarization weather radar, Adv. Geosci., 7, 127– 130, doi:10.5194/adgeo-7-127-2006, 2006b.*
*- Sebastianelli, S., Russo, F., Napolitano, F., & Baldini, L. (2013). On precipitation measurements collected by a weather radar and a rain gauge network. Natural Hazards and Earth System Science, 13(3), 605-623, doi:10.5194/nhess-13-605-2013.*
*In particular, the radar estimations reliability is assessed before and after the calibration*

*In particular, the radar estimations reliability is assessed before and after the calibration to test the effectiveness of the calibration process. Could you better clarify this aspect in the text by briefly describing the methodology followed in your work?*

A8) I'd appreciate your input on this, the sentence modified as suggested in your comment.

*Q9) Page 2, ll.13-16: I would write: "in particular, a dual polarization radar can estimate rainfall more accurately than a single polarization radar by providing reflectivity ($Z_H$), differential reflectivity ($Z_{DR}$), differential phase ($\Phi_{DP}$), specific differential phase ($K_{DP}$), and cross-correlation coefficient ($\rho_{hv}$)."*

A9) As your comments, the sentence has been revised.

*Q10) Page 2, ll.17: what did you mean with backscatter? Perhaps you meant back-scatted signal?*

A10) Yes. Backscatter is back-scatted signal from the raindrop

*Q11) Page 2, ll.17: I would delete " of hydrometeors" because for me it is redundant*

A11) The sentence has been revised.

*Q12) Page 2, ll.17: I suggest to say: "Dual-polarization radar provides characteristics of the precipitation such as precipitation type by means of… obtaining more informations about DSD (Cifelli et al., 2011), and reducing the impact of DSD variability on rainfall estimation. For these reasons rainfall estimates provided by polarimetric weather radar are better than that given by a single polarization weather radar."*

A12) The sentence modified as suggested in your comment.

*After*

"Dual-polarization radar provides characteristics of the precipitation by backscatter and differential propagation phase and therefore can reveal uncertainty in rainfall estimation resulting from DSD variability (Cifelli et al., 2011). Polarimetric parameters are sensitive to the DSD properties such as diameter, concentration, orientation, and shape. Rainfall rates derived from polarimetric radar measurements are affected by the mean shape of raindrops and canting (Brandes et al., 2002). In addition, the multi-parameters can distinguish precipitation type, and reducing the impact of DSD variability on rainfall estimation. For these reasons dual-polarization rainfall algorithms used $K_{DP}$ or combinations of $Z_H$, $Z_{DR}$, and $K_{DP}$ are better than using reflectivity factor only (Ryzhkov et al., 2005)."

*Q13) Page 2, ll.25: I would remove "of the rain" because for me it is not necessary.*

A13) The sentence modified as suggested in your comment.

*Q14) Page 2, ll.27: please split the sentence into two sentences by replacing the word "and" by a mark.*

A14) As your comments, we split the sentence

*Q15) Page 2, ll.32: I would replace "with types of storms and stages of storm development" with "with both types and stages of storm development".*

A15) The sentence modified as suggested in your comment.

*Q16) Page 3, ll.4: I would say "rainfall estimations by polarization radars are affected by errors due to different sources of uncertainties such as . . . Reviews of the different sources of uncertainties are made by different Authors in the past, for example by Zawadzki (1984), Villarini and Krajewski (2010), Sebastianelli et al., 2013, and Spina et al., 2013, which i suggest the Authors to cite:*
- Zawadzki, I.: Factors affecting the precision of radar measurements of rain, in: Proceeding of the 22d Conf. Radar Meteorology, Zurich, Switzerland, 10–13 September 1984, Amer. Meteor. Soc., 251–256, 1984.
- Villarini, G. and Krajewski,W. F.: Review of the different sources of uncertainty in single polarization radar-based estimates of rainfall, Surv. Geophys., 31, 107–129, 2010.
- Sebastianelli, S., Russo, F., Napolitano, F., & Baldini, L. (2013). On precipitation measurements collected by a weather radar and a rain gauge network. Natural Hazards and Earth System Science, 13(3), 605-623, doi:10.5194/nhess-13-605-2013.
- S. Spina, S. Sebastianelli, E. Ridolfi, F. Russo, L. Baldini, and L. Alfonso: Data selection to assess bias in rainfall radar estimates: An entropy-based method, AIP Conference Proceedings 1558, 1665 (2013); doi: 10.1063/1.4825849.

A16) The sentence modified as suggested in your comment.

*Q17) Page 3, ll.5: please delete "These measurement errors affect rainfall estimation." Because for me it's redundant.*

A17) The sentence modified as suggested in your comment.

*Q18) Page 3, ll.6: I would replace "accurate measurement and calibration of ZH and ZDR" with ZH and ZDR measurements.*

A18) "calibration of $Z_H$ and $Z_{DR}$" differ "$Z_H$ and $Z_{DR}$ measurements"

*Q19) Page 3, ll.7: I would change "accommodation" in "assessment".*

A19) The sentence modified as suggested in your comment.

*Q20) Page 3, ll.9: I would replace "measured $Z_H$ and $Z_{DR}$" in "$Z_H$ and $Z_{DR}$ measurements".*

A20) The sentence modified as suggested in your comment.

*Q21) Page 3, ll.10-11: I suggest to use the wording "vertical profile of reflectivity" instead of "measured ZH from the radar profiler (at vertical incidence)". See for example Sebastianelli et al. (2013)*

A21) The sentence modified as suggested in your comment.

*Q22) Page 3, ll.11: the disdrometer-inferred ZH*

A22) Yes.

*Q23) Page 3, ll.12: please change "by comparing reflectivity between radar and disdrometer" into "by comparison between radar and disdrometer reflectivity".*

A23) The sentence modified as suggested in your comment.

*Q24) Page 3, ll.12:* the principle has been theorized by Gorgucci et al., 1992, please insert citation.

A24) we insert Gorgucci et al. (1992)

*Q25) Page 3, ll.22-23: a mean axis ratio and a polarimetric rainfall algorithms*

A25) The sentence modified as suggested in your comment.

*Q26) Page 3, ll.24: please remove "the" and change "after" in "hereafter".*

A26) The sentence modified as suggested in your comment.

*Q27) Page 3, ll.25-26: please put into the brackets "and newly derived axis-ratio relation from 2DVD data".*

A27) The sentence modified as suggested in your comment.

*Q28) Page 3, ll.30: please remove "the" before "data"*

A28) We have modified the sentence

*Q29) Page 4, ll.2: please replace "drawn" with "given"*

A29) We have modified the sentence

*Q30) Page 4, ll.6-9: please check the English grammar and rephrase. I would write "data were used to develop a mean raindrop axis ratio relation. . .", and "The disdrometer data used in this study were collected by a 2DVD from. . ."*

A30) We have modified the sentence

*Q31) Page 4, ll.21: a frequency.*

A31) We have modified the sentence

*Q32) Page 4, ll.23: I would write "Data are obtained by using six. . ."*

A32) The sentence modified as follows.

*After*

*"The considered data are taken from six elevation angles ....."*

*Q33) Page 4, ll.29: I would say "both for Zh and ZDR calibration".*

A33) We have modified the sentence.

*Q34) Page 5, ll.1: and for rainfall estimation.*

A34) We have modified the sentence.

*Q35) Page 5, ll.2: please replace "from" with "due to".*

A35) We have modified the sentence.

*Q36) Page 5, ll.7: please write "the 2DVD rainfall estimations" instead of "rainfall calculated from 2DVD data".*

A36) We have modified the sentence.

*Q37) Page 5, ll.19-20: please rephrase. I suggest to say "In Fig. 2b and 2c we compare the axis ratio-diameter relation of Pruppacher and Beard (1970) with that found by the disdrometer before and after the correction, respectively.*

A37) We have modified the sentence.

*Q38) Page 6, ll.12-13: the sentence is poorly written, please rephrase. I suggest to write "We analyzed the rainfall cases occurred during the period . . . . Fig. 3 shows six of these cases.*

A38) The sentence modified as suggested in your comment.

*Q39) Page 6, ll.18: As the 2DVD...*

A39) We have modified the sentence.

*Q40) Page 6, ll.25: the error is with respect to rain-gauge data? Please specify.*

A40) The sentence modified as follows.

*After*

*"Therefore, the rainfall differences between 2DVD and rain gauge used in this study are limited to a maximum of 20% error, and the 2DVD data were excluded from the analysis when rainfall difference between 2DVD and rain guage was exceeding 20%."*

*Q41) Page 7, ll.1: you wrote "in suit". What did you mean? Perhaps you meant "on site"?*

*A41) the rain rate calculated from the 2DVD observation to collocated rain gauges.*

*Q42) Page 7, ll.1-3: What's the criteria you used to distinguish between the different rainfall event types? Any references? In Sebastianelli et al., 2013 you can find a description of the rainfall events types from the radar point of view. Look also the works cited in that paper about that, in particular Zhang et al., 2008, and Zhang and Qi, 2010 which define events consisting of a stratiform part and a convective one (in relation to your table 2). So I suggest to add the following references:*

*- Sebastianelli, S., Russo, F., Napolitano, F., & Baldini, L. (2013). On precipitation measurements collected by a weather radar and a rain gauge network. Natural Hazards and Earth System Science, 13(3), 605-623, doi:10.5194/nhess-13-605-2013.*

*- Zhang, J. and Qi, Y.: A real-time algorithm for the correction of brightband effects in radar-derived QPE, J. Hydrometeorol., 11, 1157–1171, 2010.*

*- Zhang, J., Langston, C., and Howard, K.: Brightband identification based on vertical profiles of reflectivity from the WSR-88D, J. Atmos. Ocean. Tech., 25, 1859–1872,2008.*

A42) We have divided rainfall event into three rainfall types such as stratiform, convective and mixed rainfall events using the radar reflectivity and one-hour rain rate calculated by rain gauge and 2DVD. We were referring to Chang et al. (2009), and rainfall rate was modified. The criterion of the rainfall events are as follows:

|  | Stratiform | Convective | Mixed(Str+Con) |
|---|---|---|---|
| Reflectivity [dBZ] | $\leq 35$ | $> 35$ | - |
| Rainfall rate[mm/hr] | $\leq 5$ | $> 5$ | - |

Chang, W. Y., Wang, T. C., and Lin, P. L.: Characteristics of the Raindrop Size Distribution and Drop Shape Relation in Typhoon Systems in the Western Pacific from the 2D Video Disdrometer and NCU C-Band polarimetric Radar, J. Atmos. Oceanic Technol., 26, 1973-1993, 2009.

*Q43) Page 7, ll.3: precipitation type.*

A43) We have modified the sentence.

*Q44) Page 7, ll.4: what did you mean with "difference rainfall"? Maybe you refer to the difference between disdrometer and rain gauge rainfall?*

A44) Yes. The word "difference rainfall" means difference between disdrometer and rain gauge rainfall

*Q45) Page 7, ll.7: I would change "larger" in "greater".*

A45) We have modified the sentence.

*Q46) Page 7, ll.17: you would write "reach about"?.*

A46) We have modified the sentence.

*Q47) Page 7, ll.18: established within.*

A47) We have modified the sentence.

*Q48) Page 7, ll.22: major and minor axis.*

A48) We have modified the sentence.

*Q49) Page 8, ll.1: 10.7 cm wavelength.*

A49) We have modified the sentence.

*Q50) Page 8, ll.2: we calculated.*

A50) We have modified the sentence.

*Q51) Page 8, ll.2: relations which.*

A51) We have modified the sentence.

*Q52) Page 8, ll.15-16: you wrote "Polarimetric rainfall relations between R and dual-polarimetric parameters are derived when rain rate is greater than 0.1 mm hr-1.". Why? to avoid the smallest raindrop diameter and therefore the particles outliers? Please clarify.*

A52) Simulated polarimetric parameters are very small when the rain rate is below 0.1 mm/hr, and to exclude the non-precipitation data, we use polarimetric parameters and rain rate data when rain rate is greater than 0.1 mm/hr.

*Q53) Page 9, ll.2: you wrote "The polarimetric radar contains systematic bias of the radar itself." It would be appropriate to introduce a bias definition. You can find a bias definition in Spina et al., 2013. In this work the calibration bias error is defined as a systematic error affecting radar estimates of rainfall independently from both the instant of measurement and the location of the sampling volume.*

A53) I'd appreciate your input on this. The sentence modified as follows.

*After*

"The radar measurements are affected by various observational errors, such as ground echoes, beam broadening and abnormal propagation echoes."

*Q54) Page 9, ll.3: I would suggest to replace "accomodation" with "assessment".*

A54) We have modified the sentence.

*Q55) Page 9, ll.4: I would suggest to write "the radar calibration" instead of "the calibration of the radar"; moreover, why the calibration was done only for light rainfall event? Maybe to avoid rain gauge error due to wind action which deflects the falling raindrops from the vertical, or it prevents the fall of the drops (updraft). So for the radar calibration you consider only stratiform cases? It is correct to say that?*

A55) Yes, you're right. We chose a continuous rainfall event for the continuity of measurement data, and we also use stable light rainfall event in order to avoid the impact of the unstable rain. In addition, to achieve accurate calibration bias of radar, the process of calibration bias should be performed in continuous and stable rainfall systems. Therefore, time of observation is short and unstable rainfall events were excluded.

*Q56) Page 9, ll.10: I would change "are point measurements" into "consist of point measurements".*

A56) We have modified the sentence.

*Q57) Page 9, ll.10: I would write "whereas radar data are measured in a sampling volume".*

A57) We have modified the sentence.

*Q58) Page 9, ll.18: I suggest the Authors to put into brackets "see Fig. 5".*

A58) We have modified the sentence.

*Q59) Page 9, ll.23: I suggest the Authors to write "… Relation derived by measurements in the wind tunnel".*

A59) We have modified the sentence.

*Q60) Page 9, ll.26-29: the sentence is too long and it is poorly written, please rephrase. I suggest to split it into two sentences. I suggest the Authors also to write "The Beard and Chuang (1987) polynomial relation (Eq. (9), black dashed line) is from 2.5 to 6.5 mm lower than the new mean axis-ratio relation values".*

A60) We have modified the sentence.

*Q61) Page 10, ll.2-3: I would write "With the exception of this case, the new axis ratio was similar to Eq. (10) for diameters ranging from 3 to 5.5 mm".*

A61) We have modified the sentence.

*Q62) Page 10, ll.5-6: please remove the sentence "This means that raindrops in South Korea are more oblate than the others"; it is my opinion that you can not assert this by comparing results of different models, which were obtained in different ways and with different data. To say something like that you should derive a mean axis-ratio relation with the same 2DVD in different part of the word.*

A62) I'd appreciate your input on this. The sentence modified as follows.

*After*

*"*These differences of raindrop shape can be caused by a variety of reasons, such as instrumental effects, fitting method, event selection, and different climatic regimes."

*Q63) Page 10, ll.12: please change "from" in "by utilizing".*

A63) We have modified the sentence.

*Q64) Page 10, ll.13: I would write "and it was compared with R derived from the 2DVD (Eq. (2))".*

A64) We have modified the sentence.

*Q65) Page 10, ll.14: the correlation coefficient. What correlation coefficient? Pearson correlation coefficient?*

A65) Yes

*Q66) Page 10, ll.15: I would write rain rate estimation as it is a disdrometer.*

A66) We have modified the sentence.

*Q67) Page 10, ll.19: observed rain rates.*

A67) We have modified the sentence.

*Q68) Page 10, ll.23: please check English grammar.*

A68) Yes

*Q69) Page 10, ll.24: I would change "when compared" in "in comparison".*

A69) We have modified the sentence.

*Q70) Page 11 ll.2: I would replace "from" with "measured by a".*

A70) We have modified the sentence.

*Q71) Page 11 ll.9: with the hourly rain rate.*

A71) We have modified the sentence.

*Q72) Page 11 ll.14: obtained by the radar.*

A72) We have modified the sentence.

*Q73) Page 11 ll.15: measured by the rain gauge.*

A73) We have modified the sentence.

*Q74) Page 11 ll.16: I would write "2DVD rainfall estimation".*

A74) We have modified the sentence.

*Q75) Page 11 ll.16: decide if you want to use showed or shown in the text.*

A75) Yes

*Q76) Page 11 ll.16: why good results. With respect to rain gauges? Please specify.*

A76) 2DVD rainfall estimation shown good results in $R(K_{DP}, Z_{DR})$, and radar rainfall estimation shown best results in $R(Z_H, Z_{DR})$.

*Q77) Page 11 ll.17: I suggest the Authors to remove the sentence part "$R(K_{DP}, Z_{DR}) > R(Z_h, Z_{DR}) > R(K_{DP}) > R(Z_h)$", and to describe the concept just in words because in this way it seems that $R(K_{DP}, Z_{DR})$ is greater than rainfall estimated by each other formula. What did you mean is that the $R(K_{DP}, Z_{DR})$ algorithm give the most reliable rainfall estimation, it's right?*

A77) As mentioned in A76), the accuracy of the rainfall estimation improved when the $K_{DP}$ parameter was used for 2DVD rainfall estimation. In the single rainfall relation with $Z_h$ (Figure 7a), an amount of scatter is present (MAE=0.95 mm h$^{-1}$ and RMSE=1.23 mm h$^{-1}$), and the scatter decrease (MAE=0.70 mm h$^{-1}$ and RMSE=0.92 mm h$^{-1}$) when the $K_{DP}$ parameter was used for 2DVD rainfall estimation (Figure 7b). These results are influenced by the variability of DSDs, and the effect of the DSD variability is declined in rainfall estimation with the $R(K_{DP})$ or $R(K_{DP}, Z_{DR})$ than that with the $R(Z_h)$. However, the accuracy of the rainfall estimation declined when the $K_{DP}$ parameter was used for radar rainfall estimation. Moreover, the radar rainfall estimations from $R(K_{DP})$ and $R(K_{DP}, Z_{DR})$ exceeded rainfall gauge measurements at lower rain rates ($\leq 5$ mm hr$^{-1}$), whereas rainfall estimations from $R(Z_h,$

$Z_{DR}$) were similar to rainfall by measured by gauges. In other words, as the rain rate increased, the uncertainty of $K_{DP}$ from the radar declined. This was because $K_{DP}$ is noisy in light rainfall. These results show that, the $R(K_{DP}, Z_{DR})$ relation is useful for heavy rainfall and $R(Z_h, Z_{DR})$ is suited for light rainfall (Ryzhkov et al., 2005).

[Figure]

Figure 7. Scatter plot of one-hour rain rate from rain gauge (RG) and BSL S-Band radar (or 2DVD) based on Eq. (4) for 18 rainfall cases: The pluses represents one-hour gauge rain rate versus radar hourly rain rate from polarimetric rainfall algorithms, and squares indicate gauge and 2DVD rain rate by different polarimetric rainfall algorithms.

*Q78) Page 11 ll.17-21: I would change "performed better" with "was more efficient"; it's not clear what did you mean when you write "on DSD statistics".*

A78) "DSD statistics" means the statistical results of the precipitation derived from the 2DVD data

*Q79) Page 12, ll.2: I would replace "declined" with "worsen".*

A79) We have modified the sentence.

*Q80) Page 12, ll.4: for lower rain rates.*

A80) We have modified the sentence.

*Q81) Page 12, ll.3-6: please delete "the radar rainfall estimations from" because it is redundant.*

A81) We have modified the sentence.

*Q82) Page 12, ll.5-6: rainfall measured by gauges.*

A82) We have modified the sentence.

*Q83) Page 12, ll.8: I would write "the uncertainty in radar estimates due to the use of KDP reduces itself".*

A83) We have modified the sentence.

*Q84) Page 12, ll.18: $Z_H$ and $Z_{DR}$ biases were calculated separately for eight rainfall events. So you calculated eight ZH biases and eight ZDR biases?*

A84) Yes. We calculated the daily $Z_H$ and $Z_{DR}$ biases, and the bias of results for each of the eight rainfall events are presented in Table 5.

*Q85) Page 12, ll.22: the BSL ZDR measurements.*

A85) We have modified the sentence.

*Q86) Page 12, ll.22-23: ZDR value is simulated by the 2DVD?*

A86) $Z_h$ and $Z_{DR}$ parameters can be simulated using T-matrix method. This method is introduction in Section 3.3.

*Q87) Page 12, ll.25: please add the word "respectively" at the end of the sentence.*

A87) We add the word.

*Q88) Page 12, ll.27: please replace the word "comparing" with the word "comparison".*

A88) We have modified the sentence.

*Q89) Page 12, ll.30: I would write ". . . obtained before and after the bias correction, respectively, while ...".*

A89) We have modified the sentence.

*Q90) Page 13, ll.1: I would write ". . . obtained before and after the bias correction, respectively, while ...".*

A90) We have modified the sentence.

*Q91) Page 13, ll.2-4: the sentence is not clear and poorly written. Please rephrase. Maybe you meant that before the bias correction the precipitation was 12.67 and after was 15.33? What's the corresponding value of the gauge?*

A91) Yes. Rain gauge recorded 15.41 mm, before the bias correction the precipitation was 12.67 mm and after was 15.33 mm. The sentence modified as suggested in your comment.

*Q92) Page 13, ll.4-5: The comparison with the rain gauge rainfall shows that after the bias correction the rainfall radar estimates were improved by about 13.71%, it's right?*

A92) Yes, you're right.

*Q93) Page 13, ll.4: the rain gauge is the ground truth. You should say that also at the beginning of the paper (introduction and abstract).*

A93) We have modified the sentence.

*Q94) Page 13, ll.7: the sentence for me is poorly written, I would say "80.12 mm respectively for rain gauge, radar before bias correction, and radar after bias correction. The radar rainfall ...".*

A94) We have modified the sentence.

*Q95) Page 13, ll.8-9: you calculated the bias for each event and then you performed a mean bias? It's right?*

A95) Yes, you're right.

*Q96) Page 13, ll.9: MAE passes from.*

A96) We have modified the sentence.

*Q97) Page 13, ll.10: decreased from.*

A97) We have modified the sentence.

*Q98) Page 13, ll.10: please add "after the bias correction" at the end of the sentence.*

A98) We add the sentence.

*Q99) Page 13, ll.11: I would write "as well as MAE and RMSE values".*

A99) We have modified the sentence.

*Q100) Page 13, ll.12: I would say "both MAE and RMSE values decrease after correcting bias, and this means that rainfall estimation tended to improve after bias correction".*

A100) We have modified the sentence.

*Q101) Page 13, ll.23-24: I would say "obtained through rainfall algorithms".*

A101) We have modified the sentence.

*Q102) Page 13, ll.24: I would say "was assessed by comparing 2DVD and BSL radar data with rain gauge measurements".*

A102) We have modified the sentence.

*Q103) Page 13, ll.25: I would change "was suited" into "is suitable".*

A103) We have modified the sentence.

*Q104) Page 13, ll.25: what's the meanings of "of the DSD statistics"? Maybe you intended "according to the DSD statistics"?*

A104) Yes.

*Q105) Page 13, ll.26: please replace "had" with "have".*

A105) We have modified the sentence.

*Q106) Page 13, ll.29: I would change "was weak" into "is noisy". It's right?*

A106) Yes, you're right.

*Q107) Page 13, ll.32: I would say "radar rainfall estimations were close to rain gauges measurements".*

A107) We have modified the sentence.

*Q108) Page 14, ll.1: I would say "different raindrops axis ratios". Did you mean the shape of the raindrops detected by the disdrometer?*

A108) We have modified the sentence.

*Q109) Page 14, ll.2: I would say "… relations, which were assessed to derive point…".*

A109) We have modified the sentence.

*Q110) Page 14, ll.3-4: I would say "obtained through rainfall algorithms".*

A110) We have modified the sentence.

*Q111) Page 14, ll.4-5: the sentence is too long and it is not clear, please rephrase. I suggest the Authors to write: "Polarimetric algorithms will be developed to obtain areal rainfall estimation. A classification of rain rate based on them will be also performed in a future work".*

A111) We have modified the sentence.

*Q112) Page 19, table 2: in the table caption please write "precipitation type" instead of "type of precipitation".*

A112) We have modified the sentence.

*Q113) Page 19, table 2: what's represents PE[%]?*

A113) *PE[%] is rainfall differences between 2DVD and rain gauge. PE[%] represented Eq. (3).*

*Q114) Page 22, table 5: I suggest the Authors to add two columns, the first one concerning the radar rainfall estimates, and the last one for rain gauge rainfall measurements.*

A114) I'll take it into account.

*Q115) Page 24, figure 2: you should clarify what represents the dotted line. Moreover you should specify that the red color correspond to the greater drop number density.*

A115) As your comments, I'll take it into account

*Q116) Page 26, figure 5: please add (b/a) at the y-label.*

A116) We have modified the figure 5.

*Q117) Page 31, figure 10: it's my opinion that it is better to replace the dotted lines with dashed lines.*

A117) We have modified the figure 10.